

# 1 Reviews and syntheses: Artisanal small-scale
# 2 gold mining (ASGM)-derived mercury
# 3 contamination in agricultural systems: what we
# 4 know and need to know

David S. McLagan[1,2,*], Excellent O. Eboigbe[1], Rachel J. Strickman[3]
[1]Dept. of Geological Sciences and Geological Engineering, Queens University, 36 Union St,
Kingston ON, Canada, K7L3N6; [2]School of Environmental Studies, Queen's University, 116
Barrie St, Kingston, ON, K7L3N6, Canada; [3]Strickman Technical Services, 1015 West Blaine
St, Seattle, WA, 98119, USA.
* Contact author: david.mclagan@queensu.ca.

## 11 Summary

ASGM is rapidly expanding and Hg-use in the sector impacts agricultural system
surrounding these spatially distributed activities. Contamination of crops from ASGM-
derived Hg occurs via both uptake from both air and soil/water. In addition to risks to human
consumers, Hg in staple crops can also be passed along to livestock/poultry further
conflating risks. Research in this area requires interdisciplinary, collaborative, and
adaptable approaches to improve our comprehension of these impacts.

## 18 Abstract

The escalating global demand for gold has fuelled the rapid expansion of artisanal and
small-scale gold mining (ASGM), which has become the largest source of mercury (Hg)
emissions worldwide. Here we synthesize current research on the pervasive contamination
of agricultural systems by ASGM-derived Hg, identifying the key environmental pathways
and subsequent risks to food security. Within these systems, Hg undergoes complex
biogeochemical transformations, with the methylation of inorganic Hg into its highly
neurotoxic form, methylmercury (MeHg), being a critical process. This is particularly
pronounced in rice paddy systems, where microbial activity and favourable redox





conditions facilitate Hg methylation, resulting in the bioaccumulation of MeHg in rice
grains—a staple food for billions. However, this synthesis reveals that atmospheric uptake
is important to total Hg loadings in rice, and more so in tissues of crops grown in unsaturated
soils. Indeed, we stress the importance of assessing all potential uptake pathways of Hg in
agricultural systems: foliar assimilation from air, uptake from soils/water (particularly MeHg
in rice), direct deposition to surfaces, and consumption of contaminated crop tissues (by
both humans and livestock/poultry), to delineate the source and ratios of the different pools
of Hg within crops and their consumers. A common shortcoming in past studies of ASGM-
derived Hg in agricultural systems is that they have commonly overlooked one or more of
these uptake pathways. These findings underscore a significant threat to global food chains
and human health through the consumption of Hg contaminated produce. Mitigating these
risks requires an improved understanding of the quantity of emissions/releases from ASGM,
input pathways, and Hg biogeochemical cycling and fate in agricultural landscapes, paving
the way for targeted interventions and sustainable management strategies to protect
vulnerable communities. We suggest that these goals can be achieved through strategic
international and interdisciplinary collaborations, novel and accessible technologies, and
care for the dissemination of scientific information to impacted communities.

## 44  1  Introduction

As a transition metal with distinctive physicochemical properties, including unique
relativistic effects, high surface tension, and liquid state at ambient temperature and
pressure, mercury (Hg) is a unique and environmentally significant element (Norby, 1991;
Jasinski, 1995; Fitzgerald and Lamborg, 2007). These unique properties have captivated
many civilizations throughout history, with Hg being used across a range of applications
including paint pigmentation, medicinal, and spiritual ceremonies (Bagley et al., 1987;
Hardy et al., 1995; Jiang et al., 2006). Use of Hg continues into the modern era particularly in
industrial, mining, and medical applications (Finster et al., 2015; Munthe et al., 2019). Hg's
recognition as a global pollutant relates to its environmental persistence, long-range
transport capabilities, and negative impacts on human and environmental health (i.e.,
neurotoxicity) (Durnford et al., 2010; Driscoll et al., 2013; Fitzgerald et al., 2007).
While all forms of Hg are toxic and we are yet to discover a biological function of the element
in the Eukarya domain at least (Peralta-Videa et al., 2009; Cozzolino et al., 2016; Grégoire
and Poulain, 2016), methyl-Hg (MeHg) is the most toxic and bioaccumulative form and the
source of the majority of Hg's impacts on human and environmental health (Clarkson et al.,



2003; Bjørklund et al., 2017). The effects of Hg (and particularly MeHg) exposure on children,
both in utero and after birth, are of particular concern due to Hg's primary toxicological
action being neurological, causing abnormalities during foetal development,
neurodevelopmental delays during childhood, with connections to autism and other mental
disabilities (Schettler, 2001; Bose-O'Reilly et al., 2010; Kern et al., 2016; dos Santos-Lima et
al., 2020). There are also links between Hg exposure and adverse effects on cardiovascular,
gastrointestinal, renal (kidneys), and pulmonary systems (Ha et al., 2017; Basu et al., 2023).
In 2013, a global treaty on Hg, the Minamata Convention, was brought into effect and signed
by 128 nations (UNEP, 2013), with the primary goal of reducing the impacts of Hg on human
and environmental health. The texts and annexes of the Minamata Convention lay out the
scientific and policy means to achieve these goals including a focus on decreasing levels of
Hg emitted to the atmosphere and released to land, water and oceans, from activities such
as artisanal small-scale gold mining (ASGM) by promoting more sustainable gold mining
practices and controlling the supply and trade of Hg (UNEP, 2013).

## 1.1 The biogeochemical cycle of mercury

Hg can exist in various oxidation states in the environment. This includes Hg(0) (elemental
or metallic), divalent or mercuric, and Hg(I) (monovalent/mercurous), although the latter is
uncommon and highly unstable in the environment and is rather a short-lived intermediary
between Hg(0) and divalent Hg (Schuster, 1991; Schroeder and Munthe, 1998). Hg(0)
dominates the atmosphere, inorganic divalent Hg (IHg(II)[i]) is the predominant form in water,
soil, and sediments, and MeHg (organic divalent Hg) is the dominant form in biota (Guzza
and La Porta, 2008; Ulrich et al., 2001; Fitzgerald et al., 2007; USEPA, 1997). IHg(II)
compounds are numerous and exhibit distinct chemical properties (i.e., $HgCl_2$ is highly
soluble, while HgS, or cinnabar, is practically insoluble) that govern their behaviour and
cycling in the environment (Schroeder and Munthe, 1998; Ulrich et al., 2001; Clarkson and
Magos, 2006; Park and Zheng, 2012; Barkay and Wagner-Döbler, 2005). While Hg is found in
a wide range of minerals, the most abundant Hg-containing minerals are cinnabar (α-HgS)
and metacinnabar (β-HgS) (Nöller, 2015).
The global distribution of Hg is achieved primarily through the atmosphere as Hg(0)
(Lindberg et al., 2007; Gworek et al., 2020), driven by its high volatility and low solubility

---

[i] We use the notation IHg(II) throughout to differentiate inorganic and organic divalent Hg (MeHg). We choose this approach over the use of IHg, as "IHg" also includes Hg(0), which has distinct physicochemical properties and behaviour from all other Hg species.



(Henry's law constant: 2.3 * 10$^{-8}$ Pa$^{-1}$; Andersson et al., 2008; Gaffney and Marleyokl, 2014),
which results in a long atmospheric lifetime of ≈4-18 months (Holmes et al., 2010; Horowitz
et al., 2017; Saiz-Lopez et al., 2018). Long-range transport via river systems also contributes,
although it is less important than the atmospheric transport pathway (Ariya et al., 2015;
Dastoor et al., 2022). Removal from the atmosphere occurs via dry deposition of Hg(0)
(dominant pathway in terrestrial systems; see Section 3 below) or oxidation to gas- or
particulate-phase IHg(II) and subsequent wet and dry deposition of these less volatile forms
(Ariya et al., 2015; Zhou et al., 2021; Dastoor et al., 2025). These depositional processes to
terrestrial and aquatic systems represent exchanges (negative fluxes), and the reverse
processes (including reduction of IHg(II) back to Hg(0) and subsequent volatilization;
positive fluxes) can also occur (Outridge et al., 2018; Dastoor et al., 2025). It is only through
burial in sedimentary materials (ocean sediments, lake sediments, and subsurface soils)
that Hg is removed from the active biogeochemical cycle (Fitzgerald and Lamborg, 2014;
Outridge et al., 2018).
IHg(II) compounds deposited, produced *in situ* from Hg(0) oxidation, or released directly into
aquatic environments such as wetlands, rivers, and lakes can undergo microbially mediated
(both enzymatic and non-enzymatic) processes that catalyse the transfer of methyl groups
from donors like methylcobalamin to IHg(II) species, forming MeHg compounds (Ullrich et
al., 2010). Methylation typically occurs under anoxic conditions in saturated sediments and
soils, but some recent studies suggest that methylation could also proceed under oxic
conditions in certain scenarios (Gallorini and Loizeau, 2021; Wang K. et al., 2021).
Representatives of sulphur-reducing bacteria, iron-reducing bacteria, methanogens,
diverse firmicutes, and other fermenting bacteria have been identified to predominantly
mediate this process in the environment (Compeau and Bartha, 1985; Lei et al., 2023). The
produced MeHg readily binds to organic matter (OM; in sediments/particles), can be taken
up by consumers, bioaccumulated, and then biomagnified up food webs (Ariya et al., 2015).
MeHg can also be demethylated biotically and abiotically (Kritee et al., 2007; Barkay and Gu,
2021). Biotic demethylation has been posited to proceeds via two pathways: (i) reductive or
mer-dependent demethylation (taxonomically widely distributed, and common in more
contaminated environments) and (ii) oxidative or mer-independent demethylation (less well
understood) (Barkay and Gu, 2021).  Abiotic demethylation occurs via direct or indirect
photolysis (Barkay and Gu, 2021).
Study of the Hg biochemical cycle has advanced significantly in the past two decades since
the development of cold-vapour introduction methods for multi-collector, inductively-



coupled plasma, mass spectrometers (MCICPMS) that has facilitated high precision
measurement and analyses of natural abundance Hg stable isotopes in samples spanning
a broad range of environmental matrices (Blum and Bergquist, 2009). There are seven stable
isotopes of Hg and significant mass-dependent (MDF; defined by δ notation) and mass-
independent fractionation (MIF; defined by Δ notation) have been observed across a broad
range of natural and anthropogenically driven processes and reactions (Bergquist and Blum,
2009; Sun R. et al., 2019; Tsui et al., 2020). Tracking Hg sources and processes with stable
isotopes analyses across time and space transcends conventional concentration analyses
by providing unique insights into the intricate behaviour and transformations of Hg across
diverse ecosystems at local, regional and global scales (Bergquist and Blum, 2009; Sun R.
et al., 2019; Tsui et al., 2020). Studies applying Hg spikes of enriched tracer isotopes
(typically in lab or heavily controlled field mesocosm experiments) have been frequently
used within the literature and are largely based on the same theoretical principles used in
natural abundance stable isotope analyses but can exploit less robust/precise
instrumentation (i.e., quadrupole ICPMS) due to the applied artificial isotope enrichments
(Hintelmann et al. 2000; Strickman and Mitchell, 2017).

## 1.2 Sources of mercury to the environment

It is important to distinguish primary emissions of Hg (predominantly to air) that augment
the mass of Hg within the active biogeochemical cycle from reemissions that represent
positive fluxes of Hg from terrestrial and aquatic matrices (i.e., vegetation, soils, water
bodies) to air, but do not alter the actively cycling mass of Hg. Reemissions more
appropriately characterize processes such as biomass burning (including wildfires) and
land use change that drive Hg back to the atmosphere as exchange process (be they
anthropogenically driven or not) rather than emissions sources (Outridge et a., 2018;
Dastoor et al., 2025). Hence, the focus of this section will be on the primary sources of Hg
emissions.
Natural primary emissions of Hg (geogenic activities and weathering of Hg-containing rocks)
are estimated at 76-300 Mg yr$^{-1}$ and make up a minor component of total annual emissions
from primary sources (Streets et al., 2019; and references therein). The most recent
inventories of primary anthropogenic emissions of Hg to air are from 2015 by Streets et al.
(2019) and Munthe et al. (2019); these sources estimate annual emissions to be 2390
(+42/-19%) Mg yr$^{-1}$ and 2220 (+27%/-10%) Mg yr$^{-1}$, respectively. In addition, Munthe et al.,



estimated 583 Mg yr⁻¹ (nonspecific uncertainty; described as large for this estimate) of Hg
are released to aquatic systems[ii].

## 1.2.1 Changing anthropogenic sources

Historically, the combustion of fossil fuels (particularly coal) has been considered the
largest anthropogenic source of mercury emissions globally (Pacyna et al., 2006; Pirrone et
al., 2010; Streets et al., 2012). The high temperatures achieved during fossil fuel combustion
liberate any residual Hg and release it as Hg(0), which typically undergoes partial oxidation
after combustion to gaseous and particulate-bound divalent Hg forms (Carpi, 1997; Pacyna
et al., 2006). More recent assessments indicate that ASGM (defined in Section 2) is now the
largest global source of anthropogenically derived Hg (Streets et al., 2019; Munthe et al.,
2019; Yoshimura et al., 2021). Munthe et al. (2019) estimate the total ASGM emissions of Hg
to air to be 838 ± 163 Mg yr⁻¹ (37.7% of total global Hg emissions to air) and total ASGM
releases of Hg to water and land to be 1221 (±637) Mg yr⁻¹. However, the authors caution that
the ASGM estimate represents a highly uncertain, "special" case scenario due to the
challenges in estimating emissions/releases from a sector with such large knowledge gaps
(Munthe et al., 2019); therefore, even these large uncertainty ranges may be
underestimates. Most ASGM Hg emissions estimates rely on a bottom-up approach based
on gold production and emission factors rather than actual Hg use (Pfeiffer and Lacerda,
1998; Seccatore et al., 2014; Streets et al., 2019; Munthe et al., 2019; Yoshimura et al.,
2021). Moreover, there is large variability not only between estimates made by different
groups, but also between different regions where ASGM occurs (Seccatore et al., 2014;
Yoshimura et al., 2021). The informal and often illegal nature of ASGM activities, which have
grown rapidly in recent decades (Wagner and Hunter, 2020; Bernet Kempers, 2020; see also
Section 2), present major challenges to Hg use inventorying (Hilson, 2008; Veiga and
Marshall, 2019).

# 2 Artisanal Small-scale Gold Mining: a "special sector"


Hentschel et al. (2002) of the International Institute for Environment and Development (IIED)
define artisanal and small-scale mining as "mining by individuals, groups, families or

---

[ii] Note the estimate of primary releases of Hg to aquatic systems does not include releases from ASGM activities as the lack of information and knowledge regarding these releases is, as yet, too large to produce a reliable estimate.



cooperatives with minimal or no mechanisation, often in the informal (illegal) sector of the
market". However, the IIED (and many other organizations and researchers) stress that a
formal definition is still lacking, and an increasing degree of mechanization and larger scale
operations are defined under artisanal small-scale mining in many jurisdictions (Hentschel
et al., 2002). This review focusses on gold mining (ASGM) alone due to the unique use of Hg
in the gold extraction process.
ASGM encompasses a wide range of techniques used to extract gold and activities range
from legal and regulated to informal to illegal activities (Veiga et al., 2006) and it contributes
≈20–30% of the world's gold production (Swain et al., 2007; Telmer and Veiga, 2009).
Estimates suggest ≈20 million individuals (including ≈3 million women and children) across
>70 countries (mainly in Africa, Asia, and South and Central America) are directly engaged
in ASGM (Seccatore et al., 2014; UNEP, 2017, Veiga and Gunson, 2020). Participant
numbers increase to at least 100 million when people indirectly dependent upon ASGM for
their livelihood are also considered (Telmer and Veiga, 2009; Veiga and Baker, 2004). The
(near) exponential growth of the ASGM sector in recent years can be attributed to soaring
gold prices, and the ease of entry into the sector and selling gold (Veiga and Hinton, 2022;
Adranyi et al., 2023). For example, the world gold spot price has increased by an order of
magnitude from ≈US\$9,000 kg$^{-1}$ in 2000 to ≈US\$105,000 kg$^{-1}$ as of 2025 (World Gold Council,
2025). For many miners, particularly those in rural communities in the Global South,
employment and survival serve as primary motivators and ASGM offers substantial financial
rewards during peak periods (Teschner, 2014; Wilson et al., 2015; Tschakert, 2009).
However, Adranyi et al. (2023) argue that these benefits come at significant social costs,
which include impacts on alternative livelihoods (i.e., loss of income for farmers as ASGM
encroaches on agricultural areas, which turns many individuals to ASGM).
The profitability of ASGM, legislative restrictions on the sector, and its proclivity to be
practiced in remote areas with less police/military presence combine to foster an
environment conducive to criminal activities led by local gangs, domestic and transnational
organized crime syndicates, and illegal armed groups (Diaz et al., 2020; Schwarz et al.,
2021). Bugmann et al. (2022) explains how industry forces are exploiting market
opportunities and coercing individuals into mining labour. Nevertheless, neither the
(il)legality nor the awareness of ASGM's impacts on human and environmental health (albeit
often limited awareness; Osei et al., 2022) have had much impact on the popularity of ASGM
or the use of Hg in the gold extraction and refinement processes (Veiga et al., 2006; Veiga
and Gunson, 2020; Thomas et al., 2019). The allure of substantial financial gains, the



scarcity of viable alternatives, and the lack of incentives for sustainable practices all
contribute to the complexity of reform within this sector (Veiga and Gunson, 2020; Telmer
and Veiga 2009).

## 2.1  The ASGM Hg amalgamation process and its impacts

Hg is used to extract gold directly from the entire mined ore (less efficient: 10-25g of Hg per
gram of gold) or from gravity ore concentrate (gold-enriched heavy fraction; more efficient:
1-3g of Hg per gram of gold) by exploiting the natural solid amalgam that forms when gold
and Hg(0) come in contact (Veiga et al., 1995; Veiga et al., 2014; Yoshimura et al., 2021). This
process produces the solid Hg-gold amalgam, tailings (waste), and residual liquid Hg, the
latter of which is reused a few times until it becomes less effective and "dirty" (inefficient),
at which point it is typically discarded into the environment (Telmer and Veiga, 2009). Once
the Hg-gold amalgam is formed (typically ≈60% gold), subsequent gold extraction is typically
accomplished by roasting of amalgam using rudimentary setups in open air, which results
in volatilization of Hg directly into the atmosphere while leaving the gold behind (Veiga and
Hinton et al., 2007; Kiefer et al., 2015; Ogola et al., 2002). This gold contains ≈2-5% residual
Hg (Veiga and Hinton, 2002) and is typically roasted a second time after purchasing by initial
gold traders (Cordy et al., 2011, 2013; Moody et al., 2020; Veiga, 2014). Although retorts
allow near complete recovery of Hg during amalgam burning, their uptake and widespread
use are limited due to costs, lack of training, and other social issues (i.e., desire to visually
observe the amalgam burning process) that are well-detailed in literature (Jønsson et al.,
2013; Hilson, 2006; Hinton et al., 2003).
Alternatives to the Hg amalgamation process do exist. These include dissolution of Hg with
nitric acid (Moreno-Brush et al., 2020; Cho et al., 2020) or the use of cyanide in place of Hg
(Marshall et al., 2020). Yet these are not popular methods due to their own inherent social,
financial, and environmental constraints (Telmer and Veiga, 2009; Brüger et al., 2018). In
addition, cyanidation is used in parallel with Hg amalgamation both to improve gold
extraction efficiencies and during transition away from Hg amalgamation (Malone et al.,
2023; da Silva and Guimarães, 2024). Concurrent use of these two methods can lead to
synergistic environmental and human health impacts as Hg-cyanide complexes are highly
toxic and increase the solubility, and hence mobility, of Hg in ASGM wastes and tailings
(Seney et al., 2020; da Silva and Guimarães, 2024). Hg amalgamation remains the preferred
method employed by ASGM to extract gold due to its simplicity, efficiency, low cost,
availability, and, ultimately, a greater confidence and trust in the Hg amalgamation process
by miners. This latter point is emphasized by the aptly titled study by Bugmann et al. (2022):





*"Doing ASGM without mercury is like trying to make omelettes without eggs": Understanding*
*the persistence of mercury use among artisanal gold miners in Burkina Faso*.
While emissions of Hg to air from ASGM activities can undergo long-range transport and
contribute to Hg's global impacts, much is deposited locally or regionally (Munthe et al.,
2019; Szponar et al., 2025). In addition, most direct releases of Hg from ASGM to terrestrial
and aquatic systems are localised (Munthe et al., 2019; Moreno-Brush et al., 2020). Hence,
communities living and working in proximity to ASGM areas are those that suffer the greatest
health impacts from this activity including the miners who can experience both inhalation
and direct dermal exposures when handling Hg(0) for gold extraction or burning amalgams
(Veiga and Baker, 2004; Bose-O'Reilly et al., 2010; Taux et al., 2022).
Another common pathway of exposure is through the ingestion of organic Hg (i.e., MeHg)
from dietary sources (Zahir et al., 2005). Fish, for instance, are exposed to MeHg both
through their environment (water) and food, with diet accounting for approximately 80-90%
of their total intake (Zahir et al., 2005). This is of particular concern for communities
impacted by ASGM activities whose major source of protein is fish (Vieira, 2006). Logically,
research on dietary exposures to Hg in ASGM affected areas is dominated by fish-focussed
studies; there are many examples of elevated concentrations of THg and/or MeHg in fish
sampled in close proximity to ASGM activities (e.g., Barocas et al., 2023; Castilhos et al.,
2015; Bose-O'Reilly et al., 2016; Maurice-Bourgoin et al., 1999). Nonetheless, fish is not the
only food consumed in regions impacted by ASGM activities.

# 3 Impacts of ASGM Hg use in agricultural regions

The surface and/or near-surface mining activities that dominate ASGM are major drivers of
land-cover change. ASGM accounts for ≈7% of deforestation in the Global South
(Hosonuma et al., 2012; Timsina et al., 2022). Additionally, the recovery of forests after
mining activities is slower when compared to other land uses (Timsina et al., 2022). ASGM
increases particle loading to rivers caused by erosion directly from ASGM activities or
indirectly after deforestation (Swenson et al., 2011; Esdaile and Chalker 2018; Moreno-
Brush et al., 2020). These issues of mining-driven deforestation and increased riverine
sediment loadings present major environmental health issues in their own rights and are the
focus of many other studies and reviews (e.g., Moreno-Brush et al., 2020; Timsina et al.,
2022; Dossou Etui et al., 2024). In addition, anthropogenically modified land-covers such as
lands used for agriculture are increasingly finding themselves in direct competition for
space with ASGM (Achina-Obeng and Aram, 2022; Adranyi et al., 2023; Yu et al., 2024;



Donkor et al., 2024). In Ghana, Achina-Obeng and Aram (2022) report that most lands
converted from agriculture to ASGM are obtained from legal sales. However, contrary
reports of ASGM "land-grabbing" also exist in Ghana and elsewhere (Gilbert and Albert,
2016; Malone et al., 2021; Adranyi et al., 2024). Indeed, conflicts between miners and
farmers/farming communities (including Indigenous Peoples) are frequent (Mestanza-
Ramón et al., 2022; Adranyi et al., 2024). A common conflict arises from the land, water and
soil degradation inflicted by ASGM that typically renders previously arable lands to be less
productive or simply infertile post mining (Gilbert and Albert, 2016; Adranyi et al., 2024).
In many areas, ASGM and agriculture continue operate alongside each other. A number of
studies cite ASGM and Hg amalgam processing occurring directly adjacent to croplands,
and farmers subsidizing their agricultural livelihood as part-time artisanal miners
(Krisnayanti et al., 2012; Mestanza-Ramón et al., 2022; Adranyi et al., 2023; 2024; Adator et
al., 2023). Hence, consumption of crops and livestock/poultry contaminated by ASGM-
derived Hg presents an additional and much less explored potential pathway of human
dietary Hg exposure (Xia et al., 2020; Sanga et al., 2023).
There are three potential pathways of Hg uptake in higher or vascular plants (the majority of
food, feed, and fuel crops are derived from vascular plants): (1) stomatal assimilation of gas-
phase Hg (0) during photosynthetic respiration, (2) surface sorption to cuticular (foliage) or
periderm (stems/bole/edible tissues) surfaces, and (3) uptake from roots (Zhou et al., 2021;
Liu et al., 2022; McLagan et al., 2022a); these processes are summarized in Figure 1. Of
these three pathways, stomatal assimilation is now considered to be the dominant
mechanism and reported to be responsible for >90% of all Hg found not only in foliage, but
all above ground plant tissues (Beauford et al., 1977; Graydon et al., 2009; Rutter et al.,
2011a; 2011b; Laacouri et al., 2013; Zhou et al., 2021; Zhou and Obrist, 2021). Moreover,
many crops are also utilized as feed for livestock and poultry. If these feedstocks are
contaminated by Hg, there is potential for accumulation in livestock/poultry and transfer to
humans after meat or animal by-product consumption. Within this section we will explore
each of these exposure mechanisms as they relate to Hg derived from ASGM and discuss
their relevancy and potential impacts on human health.




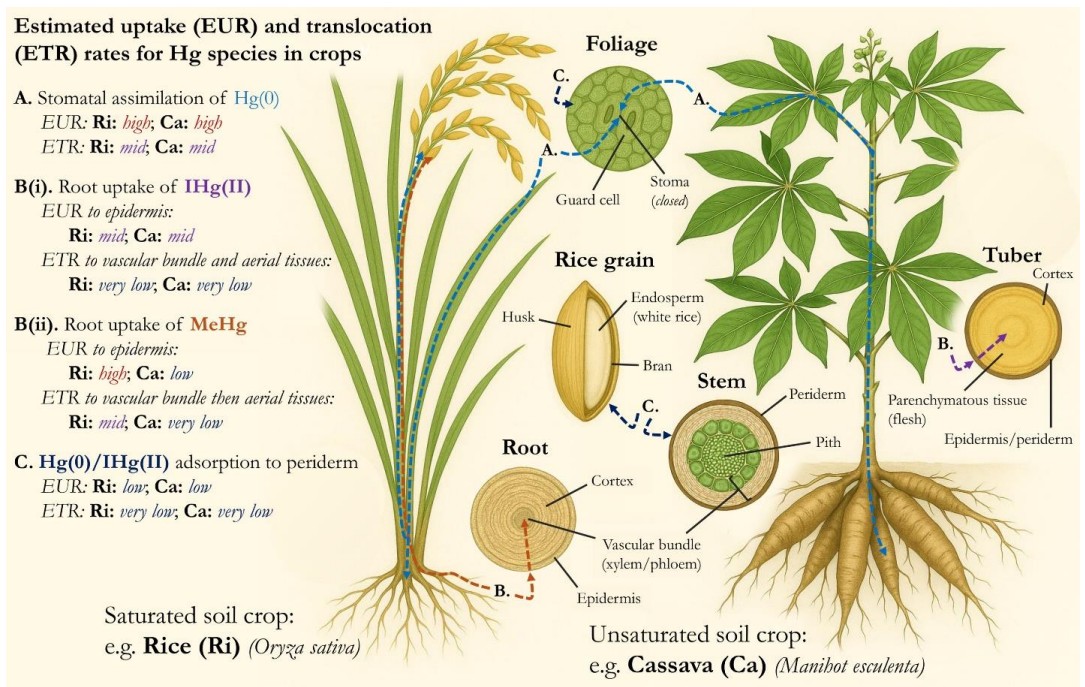


*Figure 1: Conceptual model summarizing the uptake and translocation processes of different Hg species in both saturated (i.e., rice) and unsaturated (i.e., cassava) soil crops including estimated qualitative rates based on the reviewed literature in sections 3.1 and Section 3.2. Line colours are associated with colours of species listed on the left (i.e., Hg(0) is in light blue). We note that plant and plant tissue art was developed with the purposes presentation and generic representations; hence, they may differ slightly from reality. Plant and plant tissue images were developed using ChatGPT (OpenAI), but all other parts of the figure (including labels) were constructed by co-authors.*

## 3.1 Hg uptake in crops from air: the breathers

### 3.1.1 Atmospheric Hg uptake in higher plants

Research on the uptake mechanisms of Hg from air to vegetation is highly contemporary but contains many uncertainties and knowledge gaps. The surficial sorption pathway of Hg integration into internal foliar tissue is limited largely due to the potential for Hg sorbed to the foliar cuticle to be washed off by precipitation (Rea et al., 2000; Rutter et al., 2011a; 2011b; Laacouri et al., 2013) or undergo photoreduction to Hg(0) and subsequently volatilize (Mowat et al., 2011; Laacouri et al., 2013). Dark/night experiments (when stomata are closed) have provided mixed results: some studies suggest a negative flux of Hg(0) to vegetation may occur (Converse et al., 2010; Fu et al., 2016), while other studies are less





conclusive (Fritsche et al., 2008) or indicate strong correlations between Hg(0) uptake and
stomatal conductance rates (higher uptake when stomata are open; Naharro et al., 2020).
While this suggests that a small fraction of gas-phase or surficially sorbed Hg(0) could
diffuse through the cuticle and into the internal mesophyll, this diffusion-based process is
mechanistically similar to stomatal uptake and would likely induce a similarly large,
negative (favouring lighter isotopes) fractionation of Hg stable isotopes. As such, the
discussion on atmospheric uptake pathways will focus on the stomatal assimilation
mechanism and assume all Hg within the above ground parts of plants is derived from this
uptake mechanism unless explicitly stated otherwise.
Stomatal assimilation has been directly linked to photosynthetic activity (net primary
productivity; NPP) and consequently plant growth rates (Jiskra et al., 2018; Fu et al., 2019;
Szponar et al., 2023). As such, stomatal assimilation by vegetation has been described as a
global Hg(0) pump and accounts for the largest negative flux of Hg from air to terrestrial
systems (Jiskra et al., 2018). Other factors such as stomatal conductance (itself impacted
by atmospheric/meteorological/hydrological conditions), stomatal density, photosynthetic
mechanism (i.e., C3 vs C4), cuticle thickness, cuticle roughness, plant species, and plant
and foliage life stages also influence Hg(0) uptake (Converse et al., 2010, Laacouri et al.,
2013; Wohlgemuth et al., 2020; 2021; Liu et al., 2022; Eboigbe et al., 2025). In addition, the
rate of Hg(0) foliar uptake, and consequently the THg concentration in foliage, is directly
proportional to Hg(0) concentration in air (Navrátil et al., 2017; Manceau et al., 2018; Zhou
et al., 2021), which makes the stomatal assimilation method particularly relevant in areas
with substantial Hg(0) emissions to air, including ASGM regions. Confirmation of the
dominance of the stomatal assimilation pathway and links to NPP (and other factors) has
come largely within the last 10-15 years and owes much to advancements in Hg stable
isotope research. Stomatal assimilation favours lighter isotopes and results in a MDF and
shifts in $\delta^{202}$Hg values of between -1 and -3 ‰ compared to gas-phase Hg(0) (Zhou et al.,
2021, and references therein), which creates an effective (light isotope) tracer for Hg uptake
via this mechanism in plants.
After uptake of Hg(0) into internal foliar tissue, our understanding of the processes
controlling the internal biogeochemical cycling within plants becomes somewhat less
certain. Since foliar THg concentrations increase across the growing season (Rea et al.,
2002; Laacouri et al., 2013; Wohlgemuth et al., 2020; 2021), Hg(0) must undergo oxidation
to IHg(II) (Laacouri et al., 2013; Manceau et al., 2018) to maintain the high (air) to low (within
foliage) Hg(0) concentration gradient that drives diffusion of Hg(0) into foliage. Limitations in




the interpretive power of Hg speciation analysis (McLagan et al., 2022b) restrict our
knowledge of the compounds responsible for this oxidation step, particularly at ambient
concentrations. Nonetheless, Du and Fang (1983) linked foliar Hg uptake rates to enzymatic
(catalase) activity in a high-concentration labelled isotope study, and studies using X-ray
absorption techniques on foliage samples from plants growing under highly contaminated
settings have identified Hg-thiol complexes and sulphur nanoparticles (Carrasco-Gil et al.,
2013; Manceau et al., 2021) within foliage. We require more knowledge of the biological
compounds responsible for oxidation and the resulting IHg(II) species, particularly as this
could provide critical insight into the use of vegetation in contaminated site remediation
such as at ASGM impacted areas.
As discussed, the stomatal assimilation pathway represents a net negative flux (Hg
accumulation in vegetation) overall. However, re-release of Hg(0) taken up by this pathway
has been posited to occur via photochemically-driven reduction of IHg(II) back to Hg(0) and
release back out of the stomata. Using a Hg stable isotope mass balance model, it was
estimated that ≈30% of assimilated Hg(0) is re-released from their studied species (Yuan et
al., 2018).

### 3.1.2 Translocation of Hg from foliage in higher plants

Assessments of the distribution of Hg across different plant tissues consistently indicate
foliage has the highest THg concentrations (Zhou et al., 2017; 2021; Liu, Y. et al., 2021). This
accumulation in foliage (driven by stomatal assimilation) results in litterfall representing the
major flux of Hg to soils in vegetated ecosystems (≈1000–1500 Mg yr$^{-1}$) and these same
estimates have typically also been used for as a proxy for net Hg assimilation flux into
vegetation (Wang et al., 2016; Jiskra et al., 2018; Zhou et al., 2021). Yet it has been suggested
that the use of litterfall alone likely results in a substantial underestimation of the net Hg
vegetation assimilation flux due to the translocation of Hg from foliage into other tissues
(i.e., branches, stems/boles, roots, seeds, flowers) (Zhou and Obrist, 2022). Indeed, despite
bole wood having the lowest THg concentrations of any tree tissues (Zhou et al., 2017; 2021;
Liu Y. et al., 2021), they contain the largest pool of Hg by mass of any tree tissues due to the
much greater total biomass of bole wood compared to other tissues (Liu Y. et al., 2021). Hg
storage in bole wood highlights the capacity of vegetation to translocate assimilated Hg
away from foliage.
Phloem, vascular tissue that transports solutes (i.e., nutrients, proteins, and photosynthetic
by-products such as sugars) away from the foliage within phloem sap, is suggested to be
responsible for the downward translocation of Hg (Siwik et al., 2010; Zhou et al., 2021;



Gačnik and Gustin, 2023). Throughout this downward migration, lateral translocation of Hg
from phloem, through the cambium, and into the hydroactive xylem (sapwood) must occur.
Evidence for this process lies in dendrochronological studies that (species/genus
dependent) effectively archive historical Hg(0) concentrations in tree rings (e.g., Siwik et al.,
2010, Navrátil et al., 2017, McLagan et al., 2022a; Gačnik and Gustin, 2023). Yanai et al.
(2020) and Liu et al. (2024) went further and demonstrated that this translocation from
phloem to xylem slowly reduces the amount of Hg within the phloem sap by observing a
decrease in THg concentrations in tree rings of the same age from the canopy to the ground.
Liu Y. et al. (2021) and McLagan et al. (2022a) analysed tree bark for Hg stable isotopes, and
data were highly negative in MDF ($\delta^{202}$Hg) and similar to xylem samples (tree rings) and
foliage (in the case of Liu Y. et al., 2021). This indicates foliar uptake, phloem transport, and
lateral translocation to periderm or cork (outer bark) is likely an important source of Hg in
bark (we would expect more positive MDF associated with direct deposition from air as any
such Hg would not be negatively fractionated during foliar uptake; Liu Y. et al., 2021;
McLagan et al., 2022a). From our search there have been no studies in the literature
assessing this theory in annual or bi-annual plants, such as agricultural crops.
Belowground tissues have received less attention than aboveground tissues, but Hg stable
isotope data (negative $\delta^{202}$Hg values) from trees and shrubs in a high altitude forest in China
indicated that 44-83% of Hg in roots is derived from the stomatal assimilation pathway
(Wang et al., 2020). Such data suggest root Hg storage and/or that plants could potentially
detoxify by releasing Hg taken up from air into soils. Contrary to this, isotope data from
wetland plants (i.e., rice) reflect soil isotope signatures, which is linked the uptake of
bioaccumulative MeHg that is produced under anoxic conditions prevalent in wetlands (Yin
et al., 2013). The unique case of rice, particularly in ASGM affected areas, is considered
separately in Section 3.2. We will now consider the impacts of ASGM-derived Hg
contamination in crops via stomatal assimilation.

### 428 3.1.3 Hg uptake from air in crops impacted by ASGM activities

Eboigbe et al. (2025) assessed both air and soil uptake pathways in cassava (*Manihot*
*esculenta*), peanut/groundnut (*Arachis hypogaea*), and maize (*Zea mays*) from a
contaminated (≈500m upwind) and a background (≈8km upwind) farm of a ASGM processing
site in Nasarawa State in Nigeria. Foliage was enriched 25-35x in the contaminated farm
(compared to background), and Hg stable isotope analyses revealed highly negative MDF
values in foliage ($\delta^{202}$Hg: cassava: -3.83 ± 0.15 ‰, peanut: -3.77 ± 0.27 ‰, maize: -2.51 ±
0.15 ‰), which are indicative of the negative fractionation associated with stomatal



assimilation (Eboigbe et al., 2025). Air-to-foliage enrichment factors ($\varepsilon^{202}Hg^{iii}$: -2.89 to
-1.57‰) fell into the aforementioned measured range observed in other higher vegetation
(Eboigbe et al., 2025). A two endmember Hg stable isotope mixing model based on air and
soil uptake pathways revealed 61-100% of THg in edible tubers/nuts/grains and other above
ground tissues and 26-47% of THg in roots were derived from air highlighting the dominance
of the atmospheric uptake pathway in these crops. While fraction of MeHg out of THg was
<1% (%MeHg) in all measured crop and soil samples, THg concentrations in edible parts
were above dietary guidelines and could be particularly concerning for cassava leaves (320
± 116 µg kg$^{-1}$), which are consumed in many countries (including Nigeria) (Eboigbe et al.,

445    2025).

Casagrande et al. (2020) examined ASGM-derived Hg in soy plants (*Glycine max*) and found
THg concentrations in leaves from plants grown in a ASGM affected area (mean THg: 109 ±
21 µg kg$^{-1}$) approximately three times higher than soy foliage in more background sites (THg
means: 35-40 µg kg$^{-1}$). This was despite measuring relatively low soil THg concentrations in
both ASGM (95 µg kg$^{-1}$) and non-ASGM areas (68 µg kg$^{-1}$); and indeed, THg concentrations in
other plant tissues (stems, seeds, pods, and roots) were not elevated in the ASGM affected
area (Casagrande et al., 2020). The authors link these results to atmospheric Hg uptake and
used the data to estimate a Hg deposition/accumulation rate of this ASGM affected soy farm
of 33.6 g km$^{-2}$ yr$^{-1}$ (Casagrande et al., 2020). This approach provides a novel basis for
calculating Hg accumulation from air in both background and Hg contaminated agricultural
areas. Eboigbe et al. (2025) also applied the Hg accumulation approach and calculated
fluxes of 1070±88, 98±26, 620±140 g km$^{-2}$ yr$^{-1}$ to cassava, peanuts (groundnuts), and maize
farms, respectively. These estimates include transfer to other tissues including below
ground edible parts, but Hg storage in foliage makes up the majority of Hg transferred to
crops from air (90-92%), which again raises concerns about consumption of edible foliage,
such as in cassava (Eboigbe et al., 2025).
Several other studies have assessed Hg in crops from ASGM affected areas but did not make
atmospheric Hg(0) measurements due either to logistical challenges or to the assumption
that Hg would derive largely from soil. While less ideal than paired soil and atmosphere
measurements, soil THg concentrations represent acceptable proxies for general Hg
exposure across Hg(0) contaminated areas, as deposition from air is a major source of soil

---

iii Epsilon values (i.e., $\varepsilon^{202}Hg$) are indicative of the degree of fractionation between two samples or sample matrices. For example, if $\delta^{202}Hg$ values for sample A and sample B are 1.00 and -1.00‰, respectively, then the $\varepsilon^{202}Hg$ would be -2.00‰ from A to B.



Hg, and Hg(0) in air typically correlates well with soil THg concentrations (Fantozzi et al.,
2013; Xia et al., 2020).
Golow and Adzei (2002) measured THg concentrations up to ≈35 and ≈18 µg kg$^{-1}$ in cassava
leaves and flesh, respectively, at ≈2-3 km from a mining site in Ghana; concentrations in
tissues and soils decreased with increasing distance from the ASGM site. However, these
concentrations were low compared to most other studies (Table 1). Nyanza et al. (2014)
observed THg concentrations of cassavas up to 167 µg kg$^{-1}$ in leaves, but only up to 8.3 µg
kg$^{-1}$ in flesh (little specific information relating to distance from ASGM was given). Adjorololo-
Gasokpoh et al. (2012) measured elevated THg concentrations in both cassava leaves (up
to 177 µg kg$^{-1}$) and flesh (up to 185 µg kg$^{-1}$) near another ASGM site in Ghana. While leaf THg
concentrations were again reported to decrease with distance from mining sites, there may
have been multiple sources in this study (i.e., former mines; Adjorololo-Gasokpoh et al.,
2012). A unique aspect of the Adjorololo-Gasokpoh et al. (2012) study was that they
dissected the cassava into flesh and inner and outer peels of the tuber and data from such
tissue dissection could provide critical information in discerning atmospheric and soil
uptake pathways. Nonetheless, there was little trend with distance from ASGM site in flesh,
inner peel, or outer peel (Adjorololo-Gasokpoh et al., 2012), which could be attributed to
variability in the use/emission of Hg and possible unknown sources. Our own analyses of
data from Nyanza et al. (2014; $p$ = 0.111) and Adjorololo-Gasokpoh et al. (2012; $p$ = 0.136)
indicate there was no correlation between THg concentration in cassava leaves and flesh in
these studies, which is surprising considering that stable isotope data from Eboigbe et al.
(2024) indicated the atmosphere as the source of Hg in cassava flesh.
Addai-Arhin et al. (2022) measured higher THg concentrations in both the peel (306 – 991 µg
kg$^{-1}$) and flesh (100 – 345 µg kg$^{-1}$) of cassavas at farms near (specific distance not given) at
three ASGM sites in Ghana. MeHg concentrations were measured in cassava tissues and
were <1% of THg in all samples (Addai-Arhin et al., 2022). In another study by the same
group, Addai-Arhin et al. (2021) measured both THg (and MeHg: <1.1% of THg in all samples)
in plantain (genus: *Musa*) flesh and peels at the same sites. THg concentrations in plantains
(39 – 50 µg kg$^{-1}$ in flesh and 41 – 130 µg kg$^{-1}$ in peels) were close to an order of magnitude
lower (Addai-Arhin et al., 2021) than cassava (Addai-Arhin et al., 2022) at the equivalent
farms, which highlights the species specificity of Hg uptake in crops. In the 2021 study,
much higher THg concentrations were observed in plantain flesh (mean: 580 µg kg$^{-1}$) and
peels (mean: 275 µg kg$^{-1}$) at an additional fourth farm (Odumase) adjacent to what is
[presumably] a much larger ASGM operation (Addai-Arhin et al., 2021). Interestingly, the



soils at Odumase site had lower THg concentrations than soils at other farms in their study
(Addai-Arhin et al., 2021); we speculate that the elevated THg concentration in plantain
tissues at the Odumase farms is caused by greater emissions concentrations of Hg(0) in air
from a potentially newer mine near this farm that may, as yet, not have impacted the soils
as much as has been the case at other farms (no Hg(0) measurements were taken to assess
this).
In both studies by Addai-Arhin et al. (2021; 2022) human health assessments were included
and based on USEPA daily consumption guidelines for THg in food (reference dose: 0.3 µg
of Hg per kg of body mass per day; USEPA, 2004) and estimated average daily consumption
rates (adults: 0.37 kg plantain, 0.6 kg cassava; children: 0.2 kg plantain, 0.4 kg cassava). The
Hg consumption via cassava at all farms (measured range: 0.98-3.8 µg kg$^{-1}$ day$^{-1}$; Addai-
Arhin et al., 2022) exceeded THg intake guidelines, but plantain only exceeded at the most
contaminated farm (Odumase; 3.0-3.3 µg kg$^{-1}$ day$^{-1}$; range at other farms: 0.22-0.28 µg kg$^{-1}$
day$^{-1}$; Addai-Arhin et al., 2021). While data are concerning, this may be partially offset by the
low fraction of highly toxic and bioaccumulative MeHg, all cassava and plantain samples
being below the USEPA daily MeHg consumption guideline (reference dose: 0.1 µg kg$^{-1}$ day$^{-1}$
$^{-1}$; measured: <0.026 µg kg$^{-1}$ day$^{-1}$; USEPA, 2004; Addai-Arhin et al., 2021; 2022).  A third study
by the Addai-Arhin et al. (2023) group appears to summarize these two other works, but it is
not considered for further discussion here due to their focus on cumulative peel and flesh
THg concentration data (sum of THg concentration in peels and flesh), which are not
summative data.
Sanga et al. (2023) measured THg concentrations in edible crop foliage (cassava, pumpkin:
*Cucurbita moschata*, Chinese cabbage: *Brassica rapa* subsp. *pekinensis*, and sweet potato:
*Ipomea batata*) in crop soils indicative of anomalously low Hg contamination, near
background levels (11.4±4.7 µg kg$^{-1}$), but <2km from an ASGM area in Geita Region of
Tanzania. THg concentrations were elevated and ranged from 96±14 µg kg$^{-1}$ in Chinese
cabbage to 153±128 µg kg$^{-1}$ in cassava leaves.[iv]
A similarly designed study in two villages in North Sumatra Province, Indonesia, Arrazy et al.
(2023), also measured elevated THg in foliage of cassava (mean: 2000±1600 µg kg$^{-1}$) and
katuk (*Sauropus androgynus*; mean: 4800±5900 µg kg$^{-1}$) foliage[v]; one village had dietary
intakes from these leafy vegetables (0.52-0.93 µg kg$^{-1}$ day$^{-1}$) above reference dose levels.

---

[iv] Reporting/method issues could also explain the very high crop/very low soil Hg concentration anomaly, but
we could not identify any issues from the data provided.
[v] Several other crops were studied, but each had data of only one sample and were not considered further.



However, the major difference to the Sanga et al. (2023) study was the ≈3 orders of
magnitude higher THg concentrations in crop soils (mean: 19±33 mg kg$^{-1}$). The elevated THg
concentrations in crops from both studies were hypothesized to be at least partly
associated with atmospheric uptake, though no air measurements were taken (Sanga et al.,
2023; Arrazy et al., 2023). Both studies also examined rice, discussed in Section 3.2.2.2.
A recent study in the Madre de Dios Region of Peru, examined the edible parts of six crops
(corn: *Z. mays*, rice: *O. sativa*, cassava: *M. esculenta*, plantain: *M. paradisiaca*, potato:
*Solanum tuberosum*, cocona: *Solanum sessiliflorum*) in areas deemed to be impacted by
mining (Marchese et al., 2024). Concentration levels in crops from areas listed as "impacted
by mining" were lower than in many of the previously mentioned studies ranging from 3.8 µg
kg$^{-1}$ (*n*=2) in corn to 27 µg kg$^{-1}$ (*n*=2) (Marchese et al., 2024). Even so, four of the 27 samples
exceeded maximum contaminant levels as indicated by the US Dept of Agriculture
(Marchese et al., 2024). However, these crop samples were purchased in local markets
presenting challenges in assessing distance from farms to mining sites and crop exposure
levels to Hg from either soils or air (Marchese et al., 2024). Again, rice data from this study
are interpreted in Section 3.2.2.2.
*Table 1: Information from studies of crops farmed in non-saturated soils in agricultural areas*
*impacted by ASGM activities. Tissue abbreviations: F – foliage; S – stem; R – root; T –*
*tuber/fruit; N – nut. [ ] denotes concentration.*

| Reference | Region | Country | Crop type(s) | Distance ASGM-to-Farm (km) | Farm Soil [THg] (mg kg$^{-1}$) | Farm Air [Hg(0)] (ng m$^{-3}$) | Crop tissue [THg] (µg kg$^{-1}$) | Fraction MeHg (out of [THg]) | Notes of interest |
|---|---|---|---|---|---|---|---|---|---|
| Eboigbe et al. (2025) | Nasa-rawa | Nigeria | 1. cassava 2. peanuts 3. maize | 0.5 | 76.6 ± 59.7 | 54 ± 19 | 1. F:320±116 S: 5.4±6.3 T: 0.5±10.4 R: 1.0±36.3 2. F:385±20 S: 2.1±9.9 N:6.3±21.3 R: 84.6±4.1 3. F: 82±44 S:31.7±39.3 N:1.78±1.22 R: 202±136 | <1% across all tissues for all crops and all soil samples | Crop foliage in ASGM area 25-35x THg enrichment compared to background areas. Highly negative MDF of stable Hg isotopes plant tissues (including cassava flesh) indicate uptake from air dominates over uptake from soil (N/T: 61-100%, R: 26-47% of THg derived from air). Estimated 1070±88, 98±26, 620±140 g km$^{-2}$ yr$^{-1}$ taken up by cassava, peanut, and maize. |
| Casagrande et al. (2020) | Mato Grosso | Brazil | soy | | 0.095 | NA | F: 109 ± 21 | NA | Soy foliage in ASGM area 3x THg enrichment compared to background areas. Estimated 33.6 g km$^{-2}$ yr$^{-1}$ taken up by soy. |





| Golow & Adzei (2002) | Central | Ghana | cassava | ≈2-3 | ≈100-300 | NA | F: 35 T: 18 | NA | Decreasing [THg] in soils and crop tissue with distance from ASGM |
| Nyanze et al. (2014) | Geita | Tanzania | cassava | NA | 58.4±188 | NA | F: Up to 167 T: up to 8.3 | NA | Little information on distance from site. |
| Sanga et al. (2023) | Geita | Tanzania | 1. cassava, 2. China cabbage 3. sweet potato 4. Pumpkin | <2 | 0.011±0.005 | NA | 1.F:153±128 2. F: 96±14 3. F: 117±34 4. F: 119±79 | NA | Anomalously low soil [THg] so close to ASGM. Atmospheric uptake pathway linked due to low soil [THg]. |
| Adjorololo-Gasokpoh et al. (2012) | Western | Ghana | cassava | variable | range: 94-400 | NA | Ranges: F: 93-177 T(flesh): 84-185; T(peel): 76-268 | NA | Dissected cassava tuber into peel and flesh; potentially variable ASGM sources. |
| Addai-Arhin et al. (2022) | Ashanti | Ghana | cassava | NA | range: 1290-3880 | NA | Ranges: T(flesh): 100-345; T(peel): 306-991 | <1% across all tissues | Estimated avg. daily intake was above USEPA guidelines for THg, but below for MeHg. |
| Addai-Arhin et al. (2021) | Ashanti | Ghana | plantain | NA | range: 1290-3880 | NA | Ranges: T(flesh): 33-587 T(peel): 33-292 | <1.1% across all tissues | Estimated avg. daily intake below USEPA guidelines for THg, and MeHg; exception at Odumase site with THg above guidelines. Anomalously high soil [THg]. |
| Arrazy et al. (2023) | Geita | Tanzania | 1. cassava, 2. Katuk | 0.1-0.7 | 19±33 | NA | 1. F: 2000±1600 2. F: 4800±5900 | NA | Daily Hg intake via vegetable consumption in Nauli Village above reference dose. Atmospheric and soil uptake pathways suggested. |
| Marchese et al. (2024) | Madre de Dios | Peru | assorted market crops | NA | NA | NA | 3.8-27 | NA | THg in range of crops purchased in markets of towns near ASGM activities. No information on distance from ASGM or Hg in soil or air of crops. |
| Elger et al. (2006) | Pará | Brazil | Range of crops | ≈<1km | range: 290-3840 | NA | F/S: 2600/3100 T/N:210±310 R: 410±300 | NA | Crop tissues in ASGM area ≈10-20x THg enrichment compared to background areas. |


One other study from South America (Pará State, Brazil) attempted to correlate THg in both
roots and above ground parts from a range of cultivated crops (grouped as produce) with
soil THg (no assessment of Hg(0) in air) at two ASGM impacted communities (Egler et al.,
2006). The first community appears to be village setup around a mine (we assume farms are
very close to mine) and THg concentrations were the highest measured across all studies





examining Hg in crops impacted by ASGM (mean THg concentrations: 2600 ± 3100, 210 ±
310, and 410 ± 300 µg kg⁻¹ in above ground parts, edible parts, and roots, respectively, across
all crops). At the second site (≈15 km from active ASGM sites) THg in produce was lower (120
± 110, 10 ± 10,  and 260 ± 250 µg kg⁻¹, respectively)  and only produce roots at this location
were significantly correlated with soil THg, which again suggests that atmospheric uptake is
the dominant uptake mechanism for these crops (Egler et al., 2006).
Hg concentrations in crops have been assessed in several other studies. However, these
papers lack details of sampling sites/methods and distance from ASGM (i.e., Essumang et
al., 2007), contain unclear or concerning analytical methods (Essumang et al., 2007;
Ahiamadjie et al., 2011), or had potential errors in data reporting (SSenku et al., 2023[vi]).
Therefore, these studies are not considered further.

## 3.2 Hg uptake from roots of saturated soil crops: the drinkers.

While stomatal assimilation of Hg(0) can and does occur in rice (*Oryza sativa L.*; Qin et al.,
2022; Tang et al., 2021; Aslam et al., 2022), rice is exceptional in that it also accumulates
significant amounts of Hg from the soil, due to the availability of MeHg which is formed in
the anaerobic paddy soils (Rothenberg et al., 2014). MeHg represents 40-60% of the THg
burden in rice (Rothenberg et al., 2014), which contrasts other crops that usually
accumulate only 0.05-1% MeHg even in contaminated areas (Qiu et al., 2008; Sun T. et al.,
2019; Eboigbe et al., 2025). Rice is a staple food crop for >3.5 billion people (Zhao et al.,
2020) and, globally, rice represents 10% of total MeHg intake (M. Liu et al., 2019),
emphasizing the considerable public health concerns posed by the consumption of MeHg
and IHg(II) contaminated rice.

### 3.2.1 Rice paddies: the (de)methylators

Rice paddies are characterized by cyclical flooding and drying cycles. These cycles impact
redox conditions, the forms of carbon (C), sulphur (S), iron (Fe), and manganese (Mn)
cycling, and induce strong mineral weathering (Kögel-Knabner et al., 2010). In addition, rice
paddies usually have abundant organic matter from root exudates and the reincorporation
of rice residues. The soil pool of MeHg is the dominant source of MeHg to the plant, with
multiple studies observing no evidence that *in-planta* methylation can occur (Aslam et al.,
2022; J. Liu J. et al., 2021; Strickman & Mitchell, 2017). MeHg in soil is governed by IHg(II)
bioavailability and methylation and demethylation rates, while there are multiple pathways

---

[vi] There appears to be inconsistent use of parts-per notation (ppb/ppm). Contact author did not respond to inquiries about the potential data reporting issues.



of MeHg and IHg(II) uptake into the roots and subsequent translocation into the grain,
processes described in detail below.
*3.2.1.1 Inorganic Hg (IHg(II)) bioavailability*
The rapid redox cycling created by fluctuating water conditions in rice paddies can create
"Hg species resetting" which increases the supply of soluble IHg(II) species (bio)available
for methylation (J. Liu et al., 2023; J. Wang J. et al., 2021). Logically, this supply of
(bio)available, soluble IHg(II) increases in paddies contaminated with Hg from
anthropogenic activities (including ASGM) (Ao et al., 2020; Rothenberg et al., 2014; Xu et al.,
2024). Other factors such as lower pH, oxidation of Fe(II) to Fe(III) via radial oxygen loss from
rice roots, and application of N fertilizers, can also free IHg(II) from binding sites and
increase its bioavailability for methylation (Rothenberg et al., 2014; Z. Tang et al., 2020).
*3.2.1.2 Methylation*
Mercury methylators in rice paddies appear to be dominated by iron reducers (Y.-R. Liu et
al., 2018 Z. Tang et al., 2021), methanogens (Y.-R. Liu et al., 2018, Z. Tang et al., 2021, Wu et
al., 2020), and (in some cases) sulphur reducers (Wu et al., 2020). Several aspects of the
rice paddy system influence methylation rates, with marked differences observed across
geographical and contamination gradients (J. Liu et al., 2023; Rothenberg et al., 2012).
Methylation is stimulated by the availability of labile organic carbon, which originates from
root exudates or rice straw debris (Y.-R. Liu et al., 2016; Windham-Myers et al., 2009; Zhu et
al., 2015). In addition, the draining cycle of paddies facilitates oxic regeneration sulphate
and ferric iron, electron acceptors of sulphur- and iron-reducing bacteria, as well as
promoting dissolution of iron oxyhydroxides and thus release of bound IHg(II) (Rothenberg
et al., 2014; Ullrich et al., 2001; J. Wang J. et al., 2021).
*3.2.1.3 Demethylation*
Hg demethylation in rice paddy soil has been seldomly measured, but most studies report
relatively high and consistent demethylation rate constants, suggesting resilience to
different environmental conditions (J. Liu et al., 2023; Windham-Myers et al., 2013; Zhao et
al., 2016). The taxonomic diversity of Hg demethylators may explain this, as both mer-
dependent and mer-independent demethylation have been observed in paddy soils, with
evidence for demethylation by representatives of *Clostridium spp.* (Wang J. et al., 2021),
*Catenulisporaceae*, *Frankiaceae*, *Mycobacteriaceae*, and *Thermomonosporaceae* (Y.-R.
Liu et al., 2018). Correlations between MeHg concentrations and methane emissions from
paddies suggest methanogens are important demethylators (Huang et al., 2025).
Demethylation appears to be responsive to labile organic carbon (Hamelin et al., 2015; Li &





Cai, 2012; M. Marvin-DiPasquale et al., 2000; M. C. Marvin-DiPasquale & Oremland, 1998),
but less so than methylation, based on a comparison of  methylation and demethylation in
vegetated and devegetated plots of rice paddies, which observed concomitant increases in
plant-derived labile organic carbon, MeHg concentrations, and methylation rate.
Demethylation was not measured, but any increases in this process had to have been
outpaced by the increase in methylation rate (Windham-Myers et al., 2013).
*3.2.1.4  Uptake and translocation of MeHg, IHg(II), and Hg(0) through the plant-grain system*
The uptake routes of MeHg and IHg(II) to rice differ substantially. MeHg is formed in the soil
and then absorbed through the roots; a fraction of this MeHg is retained by iron plaque or
apoplastic barriers on the root tissue, preventing complete transfer of MeHg to internal root
vascular tissues and subsequent translocation (these barriers can also prevent IHg (II)
uptake into internal tissues) (Li et al., 2015; X. Wang et al., 2014, 2015; X. B. Zhou & Li, 2019).
The review by Rothenberg et al. (2014) confirmed greater uptake of MeHg in rice by
calculating average bioaccumulation factors from previously published works of 5.5 for
MeHg and 0.32 for IHg(II). While there is uncertainty around the exact mechanisms driving
translocation, it likely occurs through conductive tissues (phloem;  xylem, (Rothenberg et
al. 2015, Hao et al., 2022; B. Meng et al., 2014; Xu et al., 2016).
Within foliage, MeHg can be photolytically demethylated via reactive oxygen species
generated by the plant itself (Li et al., 2015; Strickman & Mitchell, 2017; Xu et al., 2016). In-
planta demethylation can eliminate up to 84% of the MeHg absorbed from the soil by rice
(Tang et al. 2025) which is responsible for a protective effect valued at 30.7-84.2 billion per
year (Tang et al. 2024). Translocation of MeHg to the rice grain appears to occur in complex
with cysteine residues and concentrated in the endosperm (the "white" core of the rice
grain) (B. Meng et al., 2014).  Rice grains are referred to throughout as either unhulled, once-
milled (husk removed, bran not removed; brown rice) or twice-milled (husk and bran both
removed; white rice).  #orava
IHg(II) can also be taken up by plants in similar pathways described in Section 3.1. Sorption
of IHg(II) to roots has been observed in rice (Aslam et al., 2022; J. Liu J. et al., 2021; Strickman
& Mitchell, 2017), but similar to other crops the root epidermis likely restricts assimilation
of IHg(II) into internal root tissues limiting translocation to other tissues via this uptake
pathway. Similar to MeHg, iron plaque coatings on rice roots contribute to the root barrier
for IHg(II) via adsorption (Li et al., 2015; X. Wang et al., 2014, 2015; X. B. Zhou & Li, 2019).
Stomatal assimilation of Hg(0), subsequent oxidation, and translocation has been observed
as a source of IHg(II) to the developing rice grain (Aslam et al., 2022; J. Liu J. et al., 2021,



2021; Yin et al., 2013) as well as to the roots themselves via reverse translocation (Aslam et
al., 2022). It has also been posited that some IHg(II) could sorbed to the outer layers of the
grain (bran and aleurone layer) directly from the atmosphere (B. Meng et al., 2014).

### 3.2.2 Hg in rice impacted by ASGM activities

Globally, Hg contamination of rice in contaminated and uncontaminated areas has been
reviewed by Rothenberg et al. 2014 and Tang et al. 2020, and in Indonesia by Arrazy et al.
(2024). Our review integrates the ASGM-related body of this research with newer findings to
update our understanding of ASGM impacts on rice. We note the importance of
understanding ASGM-derived Hg contamination of rice due to prevalence of ASGM in rice
growing areas (i.e., Asia and Africa), the resulting Hg contamination of air, soils, and water,
and the presence of Hg(0), IHg(II), and MeHg in these paddy systems.
*3.2.2.1  Assessment of methylmercury production in ASGM impacted paddy systems.*
Rates of methylation and demethylation have never been estimated in ASGM environments,
and only one study has measured MeHg levels in paddy soil/sediments. Working in West
Java, Indonesia, Tomiyasu et al. (2020) measured mean MeHg concentrations of 12.3±4.8
µg kg$^{-1}$ in paddy soils ≈500 m downstream from an ASGM site compared to 6.5±2.12 µg kg$^{-1}$
in reference paddy soils ≈12 km upstream, which seems to indicate minimal differences in
methylation between ASGM and non-ASGM environments. However, accounting for the THg
concentrations in soils (0.43±0.07 mg kg$^{-1}$ and 17.4±22.5 mg kg$^{-1}$ at the reference and ASGM-
impacted paddies, respectively), %MeHg levels were highest at the reference site (1.6±>0.1
%) compared to 0.1±0.15 % at the ASGM impacted paddy (Tomiyasu et al., 2020). These
observations suggest that differences in the biogeochemical drivers of
methylation/demethylation could be more important to MeHg concentrations than THg
concentration in these systems, and that methylation was low and/or demethylation was
high at the ASGM paddy site. Predominant winds and potential atmospheric uptake of Hg(0)
could also be a factor if upstream paddies were downwind, because the speciation of Hg
could alter bioavailability for methylation, but these details were not provided.
*3.2.2.2  What do we know about methylmercury accumulation in rice in ASGM areas?*
As for other foodstuffs, the tolerable daily intake rate (the reference dose) of THg and MeHg
in rice are related to the composition of the entire diet, other MeHg sources, the duration of
exposure and the weight of the individual. While there are concerns that rice should have a
separate reference dose, because it does not offer the same beneficial micronutrients as
fish (Rothenberg et al. 2014), this work has not been undertaken. For consistency, we





therefore use the same reference doses for THg and MeHg described in Section 3.1.3 (0.3
and 0.1 µg kg⁻¹ day⁻¹ for THg and MeHg (USEPA, 2004)) for studies that discuss estimated
dietary intakes and that presented their intake calculation method. Some authors
incorporated a wet to dry correction factor to their intake calculations, which we report, if
present, since different correction factors can affect final values. alter estimates. For
studies that did not assess dietary intake, did not report their calculation method, or did not
distinguish rice from other sources of MeHg, we contextualize the health risk using the
Chinese maximum allowable concentration (MAC) for THg in rice, set at 20 µg kg⁻¹ (H. Zhao
et al., 2019). As there are no MAC values for MeHg in rice, we apply the same MAC of 20 µg
kg⁻¹ for MeHg; if the more toxic and bioaccumulative MeHg concentrations exceed this
threshold they assuredly present human and environmental health concerns. For context,
the global averages for THg and MeHg levels in rice from uncontaminated areas are 8.2 and
2.5 µg kg⁻¹ respectively (Rothenberg et al. 2014).
Information on MeHg in rice grain in ASGM areas is limited. Findings vary widely, from
minimally contaminated (1-2 µg kg⁻¹) to levels of high concern (over 100 µg kg⁻¹). These values
are within the same order of magnitude as previous findings of MeHg in rice grains from
contaminated paddies associated with other anthropogenic Hg sources (1.2-63 µg kg⁻¹,
Rothenberg et al. 2014).
Two authors employed a market-basket approach, where rice grains were purchased in
regions around ASGM activities. In addition to data on other crops (see Section 3.1.3),
Marchese et al. (2024) observed similar MeHg and THg levels in rice grain in mining-
impacted (MeHg: 7.9±7.17 µg kg⁻¹, THg: 9.1±2.9 µg kg⁻¹) compared to non-mining-impacted
areas (MeHg: 8.7±7.5 µg kg⁻¹, THg: 15.2±19.9 µg kg⁻¹). However, it was not possible to link
these market basket samples to contamination in individual mining-adjacent paddies, as
the specific growing location was unknown (Marchese et al., 2024). The same concerns
about unknown paddy locations persisted in a study by Cheng et al. (2013) in Cambodia,
who observed mean MeHg concentrations of 1.54 kg⁻¹ in market rice bought in a mining-
intensive district compared to means of 1.44 and 2.34 µg kg⁻¹ in non-mining districts. %MeHg
was not calculated for individual samples, but using overall mean THg and MeHg values, we
estimate that the %MeHg in the ASGM area was low, at ≈12%, and similar to the %MeHg
values from non-mining regions (≈20%; Cheng et al., 2013). These studies suggest that the
local commercial rice supply is relatively homogenous between mining- and non-mining
areas, which limits the effectiveness of market basket studies for determining Hg exposure
of vulnerable populations (miners and local residents) via rice in ASGM regions.





Two authors explored MeHg in rice grains derived from farms/paddies situated in close
proximity to ASGM sites. Novirsa et al. (2020) found THg concentrations (mean: 48.5 µg kg⁻¹;
range 13.8-115 µg kg⁻¹) in locally grown rice in active ASGM and farming community
(amalgamation "Hg hotspot" ≈500m from rice paddy) in Lebaksitu, Indonesia that exceeded
the Indonesian standard of 30 µg kg⁻¹ for Hg in foodstuffs; of this, 15-82% (mean: 41%) was
MeHg (mean: 14 µg kg⁻¹). Rice THg concentrations in a second village approximately 2000m
from "Hg hotspot" were lower (mean: 15.9 µg kg⁻¹; range 9.1-23.2 µg kg⁻¹), as was the MeHg
concentration (mean 9.8; range 6.5-11.7 µg kg⁻¹) but %MeHg increased (mean: 65%; range:
51-80%) (Novirsa et al., 2020). The authors intuitively link the difference in %MeHg to greater
proportional uptake of atmospherically deposited inorganic Hg (we suggest predominantly
via stomatal assimilation of Hg(0)) by rice plants grown closer to the "Hg hotspot"(Novirsa
et al., 2020). These authors estimated the probable daily intake (which incorporates an
estimate of bioavailability) of MeHg from rice and found that intake exceeded the reference
dose in the nearer village (0.139 ug/kg bw/day, range 0.079-0.199) while intake in the father
village fell below the threshold (0.063 ug/kg bw/day, range 0.040-0.093). In addition, they
found a significant correlation between hair MeHg levels and exposure via rice, indicating
that the contaminated rice was the source of the residents' MeHg intake (Novirsa et al.,
740 2020).

In their companion paper in the same area, Novirsa et al. (2019) reported very high THg
concentrations in soils at the "Hg hotspot" (32.1 mg kg⁻¹; n=1). A negative correlation
between THg concentrations and distance from source (three sites between 0.25 and 1.5
km from the hotspot) was also observed in paddy soils (from 2.26 to 0.47 µg kg⁻¹), paddy
waters (from 301 to 30 ng L⁻¹), and rice grains (from 212 to 29 µg kg⁻¹) (full details in Table 2)
(Novirsa et al., 2019). Yet they found no relationship between soil or grain THg and water THg
levels (Novirsa et al., 2019). Interestingly, this paper identified a positive correlation
between soil THg and grain THg, but the authors did not statistically relate these THg
measurements to MeHg measurements in their later work, limiting conclusions that can be
made about the relationship between THg and MeHg contamination (Novirsa et al., 2019).
Elevated MeHg concentrations were measured in rice grains (mean: 57.7±42.9 µg kg⁻¹), husk
(mean: 28.6±25.3 µg kg⁻¹), and foliage (mean: 36.0±24.9 µg kg⁻¹) from paddy fields directly
adjacent to a very highly Hg contaminated ASGM cyanidation tailings pond (mean THg in
dried solid-phase tailings: 1.63±1.13 g kg⁻¹) in Sekotong area on Lombok Island (Krisnayanti
et al. 2012). THg was not measured in rice grains, and MeHg was not measured in the tailings
ponds, making it difficult to compare estimates of methylation in soil to MeHg accumulation





in grain (Krisnayanti et al. 2012). Nonetheless, the measured mean MeHg concentration in
rice grains far exceeded the Chinese MAC of 20 µg kg$^{-1}$) (Krisnayanti et al. 2012). The very
high MeHg concentrations observed in these two studies highlight the elevated health risk
associated with consumption of rice grown in areas impacted by ASGM activities.
*3.2.2.3  What do we know about total mercury accumulation in rice in ASGM areas?*
Given that MeHg is routinely detected in rice samples when sufficiently sensitive
measurement techniques are used (Rothenberg et al. 2014), it is likely that MeHg
contamination of rice grains in ASGM areas is widespread. To help aid with comparison
between studies, we have included estimates of MeHg concentrations for all studies that
have only assessed THg in rice (those discussed in this section) by multiplying the THg
concentrations by the mean %MeHg in rice across both villages (53±12%) from Novirsa et
al. (2020) in Table 2. We emphasize that these estimates have a high uncertainty.
Concentrations of THg in rice grain have been assessed in ASGM areas of South America,
Southeast Asia, and Africa, presented in Table 2. From the studies reviewed here, THg
concentrations in rice in ASGM areas range from 1.0-1810 µg kg$^{-1}$. This range exceeds that
previously found by Rothenberg et al. (2014), who surveyed Hg in rice in control (mean 8.2
µg kg$^{-1}$, range 1.0-45 µg kg$^{-1}$) and contaminated areas (mean 65 µg kg$^{-1}$; range 2.3 - 510 µg kg$^{-1}$
$^{-1}$) impacted by Hg use in e-waste, cement production, and other industrial and mining
activities, including some earlier studies on Hg in rice in ASGM areas. The literature
summarized below excludes studies covered in the Methylmercury section (3.2.2.2.1),
which includes the only work from South America (Marchese et al. 2024). In addition, several
studies were excluded due to issues with quality control reporting or inconsistencies in data
tabulation in text (Hindersah et al 2018, Ramlan et al. 2022, Saragih et al. 2021, Ssenku

780  2023).

**Southeast Asia**, particularly Indonesia, has received more attention than other regions, but
levels of THg contamination were variable and did not always translate to elevated THg in
rice. For instance, surprisingly low rice THg contamination was observed by Appleton et al.
(2006), who studied  Hg in waters, sediments, different types of agricultural soils, mussels,
fish, bananas, and rice prepared in various ways in an irrigated farming area in the Naboc
watershed, downstream of an ASGM site on Mindanao Island, the Philippines. Expectedly,
irrigation of rice paddies with Hg-contaminated water from the mine resulted in significantly
higher THg concentrations in paddy soils (mean: 24, range 0.05-96 mg kg$^{-1}$) compared to
unirrigated banana and corn field soils (means of 0.12 and 0.27 mg kg$^{-1}$ respectively)
(Appleton et al., 2006). However, rice Hg levels ranged from an average of 20 µg kg$^{-1}$ for once-



milled rice (range 1-43 µg kg$^{-1}$), 18 µg kg$^{-1}$ for twice-milled rice (range 8-50 µg kg$^{-1}$) and 15 µg
kg$^{-1}$ for cooked twice-milled rice (range 6-37 µg kg$^{-1}$) (Appleton et al., 2006). These results
highlight that the preparation method of rice, including cooking, has the potential to
modulate exposure risk. The authors suggested that the surprisingly low THg concentrations
in rice, given the degree of soil contamination, could be the result of the post-harvest
sampling strategy, which combined rice grown in paddies with variable magnitudes of
contamination (Appleton et al., 2006).
In contrast, Pataranawat et al. (2007) conducted THg measurements of paddy waters, soils
and rice (as well as other matrices) around an ASGM facility in Phichit Province, Thailand,
and observed that once-milled rice had very high THg concentrations (228±55 µg kg$^{-1}$).
However, the surface soil THg concentrations (unclear if this was paddy soil but associated
with the rice samples: 120±80 µg kg$^{-1}$) were lower compared to other ASGM sites (Table 2)
(Pataranawat et al., 2007). The authors also measured elevated Hg dry deposition rates in
the area (range: 24-139 µg m$^2$ day$^{-1}$; compared to background dry deposition rates in Japan:
8.0 ± 2.7 µg m$^2$ day$^{-1}$; Sakata and Marumoto, 2005) and suggested stomatal assimilation of
Hg as the explanation for the elevated rice and low paddy soil THg concentrations. However,
the study lacked both MeHg measurements in rice or paddy soils (a significant fraction of
the THg content of rice), and foliage Hg measurements to more comprehensively assess this
hypothesis (Pataranawat et al., 2007).
Working in three villages within 15 km (specific distance of each village to ASGM site not
listed) of an active ASGM site in North Gorontalo Province, Indonesia, Mallongi et al. (2014)
observed very high THg concentrations in both once-milled (up to 1812 µg kg$^{-1}$) and twice-
milled rice (up to 1080 µg kg$^{-1}$) (Table 2). Stomatal assimilation was again speculated as a
potential contributor to the high THg concentrations in rice due to high measured dry
deposition rates (166 – 219 µg m$^2$ day$^{-1}$) but the authors again lacked the appropriate
analyses to confirm this (Mallongi et al., 2014). They also included a diet-based health
assessment that raised concerns of residents consuming this rice in this area, particularly
brown rice from the village closest the ASGM site (Mallongi et al., 2014).
Giron et al. (2017) surveyed the soil and rice grain THg concentrations on Masbete Island,
the Philippines, at rice fields near an ASGM site, and a reference site ≈37 km away. They
found that paddy soil THg concentrations were extremely elevated in the ASGM site (6880-
7810 µg kg$^{-1}$) compared to the distant region (13-74 µg kg$^{-1}$). Unhulled and once-milled rice
concentrations were also elevated at the ASGM site in comparison to the control site (117-



133 and 1.6-13 µg kg$^{-1}$, respectively; Giron et al., 2017). The ASGM site was directly adjacent
to a tailings pond and reportedly received tailings contaminated water (Giron et al., 2017).
Arrazy et al. (2023) measured somewhat lower THg concentrations in rice (mean: 50±33 µg
kg$^{-1}$) from similarly contaminated ASGM-derived Hg paddy soils (mean THg: 5600±12000 µg
kg$^{-1}$) in rice-growing villages with active amalgamation and amalgam burning North Sumatra
Province, Indonesia. In this study, THg concentrations in rice were correlated with THg in
soils and distance from amalgam burning sources, but all rice sources were 300-600m from
these sites; hence all sites were heavily contaminated (Arrazy et al., 2023). The authors also
calculated average daily intake values of THg from rice for adults (0.30-0.34 µg kg$^{-1}$ day$^{-1}$) and
children (0.54-0.63 µg kg$^{-1}$ day$^{-1}$) using a wet/dry conversion factor set at 0.91; both adults
and children had exposures above the USEPA reference dose level (Arrazy et al., 2023).
A small epidemiological study exploring the health effects of mercury exposure in an ASGM
village in Indonesia observed that the local rice supply, upon which the villagers depended
entirely, was highly contaminated (68-1186 µg kg$^{-1}$ of THg in unhusked, once-milled, and
twice milled stored rice of various ages; mean value 301 µg kg$^{-1}$), and estimated THg intake
rates of 0.14 µg kg$^{-1}$day$^{-1}$ for adults and 0.57 µg kg$^{-1}$day$^{-1}$ for children (Bose-O'Reilly et al.,
2016). Of the 18 villagers examined, 15 were experiencing symptoms of clinical Hg
intoxication (Bose-O'Reilly et al., 2016). These affected individuals had relatively high THg
levels in hair combined with relatively low THg levels in urine, which is indicative of the
manifestations of MeHg exposure rather than inorganic Hg exposure (Bose-O'Reilly et al.,

844  2016).

**Shifting to Africa,** studies of ASGM impacted rice paddy systems were typically indicative
of lower concentrations of THg in paddy soils compared to studies in SE Asia. This may
reflect more distributed cultivation of rice in Africa, greater competition for the same land
resources in SE Asia, simply that researchers have not been able to study more heavily
impacted rice paddies in Africa due to social/geopolitical drivers or funding/capacity issues.
Taylor et al. (2005) explored Hg in rice around a mining area in Nigeria using a market basket
approach combined with a single paired rice-soil sample as part of a more complex survey
of dietary metal contamination across multiple environmental compartments. They found
that rice grown within 5 km of the ASGM site had THg concentrations of 31-35 µg kg$^{-1}$ and Hg
in these paddy soils had a mean THg concentration of 120 µg kg$^{-1}$ (Taylor et al., 2005).
However, other paddies that were not sampled for rice had much higher THg levels (up to
5100 µg kg$^{-1}$) (Taylor et al., 2005); hence, the measured THg concentrations of rice may be
on the low end of actual rice concentrations in this ASGM affected area.



Kinimo et al. (2021) assessed Hg contamination of rice and human exposure at two ASGM
sites in rice-subsistence communities of Ababou and Bonikro, in south-central Cote
d'Ivoire. In once-milled rice, THg concentrations were 20±10 µg kg$^{-1}$ at Bonikro (53% of
samples exceeded Chinese MAC threshold), and 40±20 µg kg$^{-1}$ in Agabou (all samples
exceeded) (Kinimo et al., 2021). Nonetheless, calculated average daily intakes of Hg via rice
fell below the USEPA threshold (Bonikro: 0.0075 µg$^{-1}$ kg$^{-1}$ day$^{-1}$, range 0.0029-0.016; Agabou
mean 0.018 µg$^{-1}$ kg$^{-1}$ day$^{-1}$, range 0.0073-0.079). However, their wet/dry conversion factor
was set to 0.085, an order of magnitude lower than that used by other authors here (Arazzy
et al., 2023: 0.91, Sanga et al., 2023: 0.86) and may have biased these estimates (Kinimo et
al., 2021).
Finally, Sanga et al. (2023), measured elevated rice grain THg concentrations (mean:
97.6±34.3 µg kg$^{-1}$) near (<2 km) an ASGM site in Geita Region of Tanzania and calculated a
daily intake of Hg from rice of 0.429 µg$^{-1}$ kg$^{-1}$ day$^{-1}$ using a wet/dry conversion factor of 0.86;
both rice concentrations and intake rates exceed safe thresholds. Sanga et al. (2023)
observed that rice grain THg concentrations (mean: 75.6±0.005 µg kg$^{-1}$) at a "background"
site (≈9 km away) were also above the Chinese MAC (EDIs not estimated at this site). Despite
the elevated Hg concentration in rice grains, paddy soil THg concentrations at both the near
mining (mean: 32.1±38.2 µg kg$^{-1}$) and "background" (mean: 10.6±2.3 µg kg$^{-1}$) were
anomalously low and near background levels (Sanga et al., 2023). Atmospheric foliar uptake
is briefly discussed with relation to other crops examined in this study but not linked directly
to the observed high rice Hg and low soil Hg data (Sanga et al., 2023). We posit that foliar
uptake and translocation of IHg(II) to rice grains could drive this discrepancy. [vii]

---

[vii] Reporting/method issues could also explain the very high rice/very low soil Hg concentration anomaly, but
we could not identify any issues from the data provided (the same anomaly was noted for other crops in this
study; footnote iv).





*Table 2: Summary of studies examining Hg in rice. All data presented in means ± standard deviation if these data were available or could be calculated from tabulated datasets. If means ± standard deviations were not provided or could not be calculated, we provide the values supplied by the authors (means and ranges, mean only, or ranges only). For studies without measurements of rice grain MeHg, we have provided a coarse estimate of the MeHg content based on the rice grain THg values and the average %-MeHg value observed by Novirsa et al. 2020 (53%).*

| Reference | Research type | Region | Country | Rice preparation type | Dist. ASGM-to-site (km) | Sub-site description | Farm Soil [THg] (mg kg⁻¹) | Farm Soil [MeHg] (µg kg⁻¹) | Rice Grain [THg] (µg kg⁻¹) | Rice Grain [MeHg] (µg kg⁻¹) | %MeHg (out of [THg]) | Notes of interest |
|---|---|---|---|---|---|---|---|---|---|---|---|---|
| Arrazy et al. (2023) | field study | North Sumatra | Indonesia | once milled | 0.1-0.25 | | 5.6±12 | NA | 50 ± 33 | 27 * | NA | |
| Marchese et al. (2024) | market basket | Madre de Dios | Peru | Un-stated | Un-stated | mined regions | NA | NA | 9.1±2.9 | 7.9±7.1 | 99±50 | |
| | | | | | | unmined regions | NA | NA | 15.2±19 | 8.7±7.4 | 88±60 | |
| Sanga et al. (2023) | field study | Geita District | Tanzania | unstated | | 0-2 km from mining | 0.032 ±0.038 | NA | 97.6±34.3 (75.2-158.7) | 52 * | NA | |
| | | | | | | >9 km from mining | 0.0106 ±0.0035 | NA | 75.6±0.4 (75.2-75.9) | 40 * | NA | |
| Kinimo et al. (2021) | field study | South-Central Region | Cote d'Ivoire | Unclear if once or twice milled | 0.1-3 | Agabou | NA | NA | 20-160 | 10.6-84.8 * | NA | 53% of samples exceeded 20 ng/g |
| | | | | | 0.1-3 | Bonikra | NA | NA | 10-30 | 5.3-15.9 * | NA | |
| Tomiyasu et al. (2020) | paddy soil only | West Java | Indonesia | NA | 0.1-2 | paddies near ASGM sites | 17.4 ±22.5 | 12.3 ± 4.8 | NA | NA | 1.6±0.1 | Values tabulated from supplementary data. Snail MeHg and THg were measured but not rice. |
| | | | | | 10 | paddies in a national park | 0.43 ±0.07 | 6.5 ± 2.1 | NA | NA | 0.0015 | |
| Novirsa et al. (2020) | survey of home rice supplies | West Java | Indonesia | Unclear if once or twice milled | 0.5-2 | village adjacent to mine | NA | NA | 48 (13.8-115) | 14.0 (4.9-20.7) | 41 (15-82) | Hyperlocal rice cultivation confirmed in survey data; 97% of residents grew own rice near homes or bought it from neighbours. |
| | | | | | | village 2km from mine | NA | NA | 15.9 (9.1-23.2) | 9.8 (6.5-11.7) | 65 (51-80) | |
| Novirsa et al. (2019) | field study | West Java | Indonesia | once milled | 0.5-1.5 | paddy 0.25 km from ball mill and mining area | Soil: 2.26±0.15 water: 301 ±420 ng/L | NA | 211 ± 11 | 112 * | NA | |
| | | | | | | paddy 0.5-1km from ASGM sites | Soil: 0.63±0.34 Water: 66± 100 ng/L | NA | 91 ± 13 | 48 * | NA | |
| | | | | | | paddy 1-1.5 km from ASGM sites | Soil: 0.47±0.12 Water: 30 ±31 ng/L | NA | 29± 1 | 15 * | NA | |
| Giron et al. (2017) | field study | Masbate Island | Philippines | unhulled and once milled | 0.5-1 | ASGM mining district | 6.888-7.812 | NA | Unhulled: 117 1x milled: 133 | Unhulled: 62 * 1x milled: 71 * | NA | Mean values only reported, no estimates of variance/uncertainty |
| | | | | | ~37 | non-ASGM district | 0.013-0.074 | NA | Unhulled: 1.6 1x milled: 13.1 | Unhulled: 0.8 * 1x milled: 6.8 * | NA | |




| Reference | Study type | Location | Country | Milling | Distance | Site | | | | | | Comments |
|---|---|---|---|---|---|---|---|---|---|---|---|---|
| Bose-O'Reilly et al. (2016) | field study | West Java | Indonesia | unhulled, once milled, and twice milled | not reported | | NA | NA | 310 (68-1186) | 164 * | NA | Local ASGM-impacted rice consumed by community. Stored rice of variable ages & types. Paddies irrigated with Hg contaminated water, paddy-ASGM distances not reported. |
| Mallongi et al. (2014) | field study | Goront-alo Prov. | Indonesia | Once and twice milled | within 15 km radius | Wubudu | 1.52-3.58 | NA | 1x mill: 1042-1821  2x mill: 603-1084 | 1x mill: 552-965*  2x mill: 320-575* | NA | |
| | | | | | | Motihamulo | 0.48-2.9 | NA | 1x mill: 795-915  2x mill: 628-754 | 1x mill: 421-485*  2x mill: 332-400* | NA | |
| | | | | | | Dulukapa | 0.88-2.26 | NA | 1x mill: 122-254  2x mill: 113-183 | 1x mill: 65-135 *  2x mill: 60-97 * | NA | |
| Cheng et al. (2013) | market basket | Kratie Region | Cambodia | not stated; likely twice milled | not stated | ASGM mining district | NA | NA | 12.7 (9.90-16.7) | 1.54 (1.06-2.31) | 12 | %-MeHg values were calculated from mean MeHg and THg values |
| | | Kamp-ng Cham Region | | | | non-mining district | NA | NA | 8.14 (6.16-11.7) | 1.44 (1.17-1.96) | 18 | |
| | | Kandal Region | | | | non-mining district | NA | NA | 10.21 (5.91-15.1) | 2.34 (0.48-5.23) | 23 | |
| Krisnaya-nti et al. (2012) | field study | Lombo-k Island | Indonesia | one milled | field directly adjacent to cyanidation tailings pond | | not measured; THg in solid-phase tailings of adjacent pond was 1630±1130 | NA | NA | grain: 57.7±42.9 hull: 28.6±25.3 leaf: 36.0±24.9 | NA | Maximum grain MeHg concentration of 115 µg kg-1 |
| Patarana-wat et al. (2007) | field study | Phicit Prov. | Thailand | once milled | 1-6 | | 0.12±0.8 | NA | 228±55 | 121 * | NA | All samples, even those further from the mine, far exceeded the maximum allowable concentration. |
| Appleton et al. (2006) | field study | Minda-nao Island | Philippines | once milled | 10 | | 24 (0.05-96) | NA | 20 (1-43) | 11 * | NA | Rice from storage, soils from adjacent paddies receiving ASGM contaminated irrigation & silt tailings |
| | | | | twice milled | | | | NA | 18 (8-50) | 10 * | NA | |
| | | | | Twice milled cooked | | | | NA | 15 (6-37) | 8 * | NA | |
| Taylor et al. (2005) | market basket | Geita District | Tanzania | unhulled | <5 km | | 0.3 (0.005-5.1) | NA | 31-35 | 17-19 * | NA | Market based, but reported to be within 5 km of ASGM site |


The literature summarized in this section suggest that both uptake through roots (likely of
MeHg) and Hg(0) uptake through foliage are important determinants of grain THg
concentrations in rice grains in ASGM areas. This conclusion is largely derived from the data
inconsistencies between THg concentrations in paddy soils (and on occasion also distance
from source) and THg concentrations in rice, which indicate that simple soil THg
concentration was not the only control on grain THg concentration in grains (i.e., Appleton





et al., 2006; Pataranawat et al., 2007; Sanga et al., 2023),  as well as the comprehensively
structured study by Aslam et al. (2022) which strongly suggested an atmospheric route of
Hg(0) uptake is occurring in rice.  This does not discount the importance of uptake from roots
in ASGM areas, as there are studies that have observed a positive rice grain – paddy soil THg
correlation (i.e., Arrazy et al., 2023; Novirsa et al., 2019). While the authors interpreted this
to mean that the soil was the source of grain THg,  we believe it is more likely to be the result
of bioaccumulation of the (unmeasured) methylated fraction of the total Hg pool, given that
MeHg is readily detected in rice grains at high levels in ASGM areas (Krisnayanti et al., 2012;
Novirsa et al., 2020, Rothenberg et al. 2014). While we cannot fully discount the possibility
of direct soil uptake of IHg, the presence of IHg in rice grain could also be explained by the
recently confirmed in-planta demethylation pathway (Tang et al. 2024), or stomatal uptake
and subsequent reverse translocation (Aslam et al 2022) followed by loading to the
developing grain. Studies to better understand the local controls over both uptake
mechanisms, and why anomalously low rice Hg occurs in areas with high paddy soil Hg (and
*vice versa*), should be the focus of future research

## 3.3 Hg uptake by livestock/poultry: the consumers

Restricting our definition of agriculture to more traditional terrestrial farming practices (fungi
or aquaculture farming are not considered), we must also consider potential Hg exposures
through the consumption of Hg contaminated livestock, poultry, or their egg/dairy by-
products; yet research in this area is very limited. Hg in herbivorous, mammalian livestock
(i.e., cattle, sheep) and their milk is suggested to be derived largely from Hg in feedstocks
with inhalation deemed a minor uptake pathway (Verman et al., 1986; Crout et al., 2004;
Parsaei et al., 2018). Qian et al. (2021) mention that Hg speciation, and specifically the
fraction of MeHg in the contaminated feedstocks is likely to impact the extent of
bioaccumulation in poultry and livestock. Yet the authors did not directly measure any form
of Hg in the animals or animal products (only THg and MeHg in plants) and simply highlight
this potential exposure pathway (Qian et al., 2021).
Verman et al. (1986) demonstrated that dosing cattle (*Bos taurus*) for three months with
feedstocks enriched in inorganic Hg (1.2 – 3.1 mg of Hg per day) above control doses (0.2 mg
of Hg per day) can result in accumulation of Hg in the animals, particularly in the liver (9x Hg
enrichment in liver tissue vs control) and kidneys (16x Hg enrichment in kidney tissue vs
control). Similar results (Hg enrichment in kidneys and liver compared to muscle) were
found by Crout et al. (2004) by dosing cattle feedstocks with isotopically labelled inorganic
Hg, but no control cattle were used in this study. These data present livestock health





implications due to the known impacts of Hg on the gastrointestinal and renal systems in
humans and other mammals (Ha et al., 2017; Basu et al., 2023). Indeed, data demonstrating
the concentration of Hg in the kidneys and liver of terrestrially farmed animals not only stress
the need for caution/avoidance of human consumption of these tissues in regions with
known Hg pollution issues such as ASGM areas, but they also highlight renal and
gastrointestinal health risks in humans consuming of crops contaminated by inorganic Hg
(via the stomatal assimilation pathway).

### 3.3.1  Hg in terrestrially farmed animals impacted by ASGM activities

Basri et al. (2017) measured significantly higher THg concentrations in hair of cattle living
inside (<2 km from; 11.4 ± 9.5 mg kg$^{-1}$) compared to outside (>8 km from; 2.9 ± 2.5 mg kg$^{-1}$)
an ASGM area on the island of Sulawesi. THg concentrations in hair also increased with
cattle age, which suggests Hg is bioaccumulating the cattle (Basri et al., 2017). In a follow-
up study of the same area, the authors examined soils and forage grasses (*Imperata*
*cylindrica*, *Megathyrsus maximus*, and *Manihot utilissima*) that these cattle feed upon;
though THg concentrations in soils were significantly higher inside compared to outside the
mining area, the difference for forage grasses (inside vs outside) was not determined to be
significant (Basri et al., 2020).
A study from Ghana examined liver, kidney, and muscle in sheep (*Ovis aries*), goat (*Capra*
*hircus*), and chicken (*Gallus gallus domesticus*) and in each case THg concentrations were
greater in kidneys (7 ± 8, 3 ± 2, and 12 ± 8 µg kg$^{-1}$, respectively) than liver (3 ± 3, 1 ± 1, and 11
± 7 µg kg$^{-1}$, respectively), which were higher again than muscle (non-detect, non-detect, and
1 ± 1 µg kg$^{-1}$, respectively) (Bortey-Sam et al., 2015). While the study did use a robust and
highly sensitive THg analyser (MA3000, NIC), it appears low sample mass impacted the
detectable THg concentration in the results (Bortey-Sam et al., 2015). Furthermore,
chickens were market bought, and sheep and goat were obtained from slaughterhouses;
hence, little specific information on feed and exposures could be determined (Bortey-Sam
et al., 2015).
Marchese et al. (2024) assessed THg in feathers, eggs, and internal tissues (muscles and
organs) and MeHg in eggs and internal tissues of "backyard" chickens from an ASGM
community and an upstream remote community in the Peruvian Amazon (Madre de Dios
Region). Median THg concentrations were 7.3x higher muscle and organ tissues and 3.6x
higher in feathers from mining areas compared to the background site; there was no
significant difference in egg THg or MeHg between the sites (Marchese et al., 2024).
Interestingly, chicken livers had the highest THg concentration, but lowest fraction of MeHg



(54%; MeHg fraction was up to 100% in other tissues: spleen and back muscle) and MeHg
fractions were significantly lower in ASGM area than background (Marchese et al., 2024).
The omnivorous nature of chickens and other poultry presents additional dietary variables
to their own and subsequent human (via consumption of meat and eggs) exposures to Hg;
their diets can vary greatly depending on how they are reared (Klasing 2005). Indeed,
Marchese et al. (2024) observed significantly higher $\delta^{13}C$ data in chicken feathers in
background area compared to ASGM area, suggesting differences in chicken diets between
the sites. The lack of difference in $\delta^{15}N$ between the sites indicates that this is not associated
with a significant change in trophic feeding level but rather changes in plant food types
(Marchese et al., 2024). Despite these differences authors conclude that differences in
environmental exposure levels drive the observed differences in chicken THg and MeHg
concentrations at the ASGM and background sites (Marchese et al., 2024). In addition to Hg
in chicken and crops, the Marchese et al. (2024) study also examined Hg in fish and
combined all these data to produce probable weekly Hg intake values for humans in these
regions. As expected, fish are the dominant dietary source of Hg make up ≈82% of THg intake
(≈96% of MeHg) compared to ≈17% (≈3%) and ≈1% (≈1%) for crops and chicken, respectively
(Marchese et al., 2024). Although the high THg concentration and lower MeHg fractions
observed in chicken tissues (particularly livers) again raises some concern of inorganic Hg
contamination and potential bioaccumulation in (particularly in detoxifying organs of)
poultry/livestock in ASGM affected areas, the much larger Hg burden from fish consumption
adds crucial perspective to dietary concerns relating to poultry/livestock consumption at
least based on results of the Marchese et al. (2024) study.
Two other studies have examined THg concentrations in poultry blood. Abdulmalik et al.
(2022) measured significantly higher THg blood concentrations (0.08–0.09 µg L$^{-1}$) in chickens
sampled within 1 km of ASGM compared to control chickens (non-detectable
concentrations). While Aendo et al. (2022) measured much higher THg concentrations in
poultry blood (mean THg range: 20–43 µg L$^{-1}$), linkages between concentrations and
proximity to mining were less clear. Only free-grazing ducks (specific species not listed)
within a mining area (albeit a large area, within 25km radius, deemed to be impacted by
mining) had significantly high THg concentrations to those outside the mining area; chickens
and farmed ducks were not significantly different (Aendo et al., 2022).



## 4 Implications and future research direction

The global extent and rapid growth of ASGM places critical emphasis on the need to address the serious environmental and human health risks presented by ASGM Hg use. Ideally, such efforts should start with improving our understanding of Hg emissions and releases associated with ASGM, which are highly uncertain and currently based on poorly constrained knowledge of Hg use, gold production, and the sheer scale of the rapidly growing and largely informal/illegal sector. The implementation of accessible, low-cost, low-tech solutions such as the Hg passive air sampler method utilized by Szponar et al. (2025) to assess Hg(0) concentrations, exposures, and emissions to air from ASGM activities are needed to generate the robust monitoring data needed to better assess ASGM Hg emissions and releases. Efforts to model ASGM emissions and fate remain hampered by our limited knowledge of Hg use inventories. Nonetheless, novel ASGM Hg modelling efforts that account for the importance of the sink of Hg to terrestrial vegetation (particularly in the more heavily vegetated tropics where much ASGM occurs) such as that presented by Hedgecock et al. (2024) will undoubtedly improve our understanding of the cross-compartmental distribution and air-vegetation dynamics of Hg in ASGM areas. Considering >55% of the planet's ice-free land has been converted to farming or lands for human settlement (Ellis et al., 2010), it could be beneficial to adapt such models to include agricultural biomes.

There have been considerable advancements, paradigm shifts even, in terms of our understanding of the importance of Hg(0) uptake (stomatal assimilation) by plants from the atmosphere, now understood to be the dominant flux of Hg from air to terrestrial systems. However, there needs to be a greater focus on such research from the context of ASGM and agricultural crops. The recent work by Eboigbe et al. (2025) using Hg stable isotopes analyses of soils, air, and different crop tissues provided critical insight into the importance of the stomatal assimilation pathway in staple crops. While many previous studies of Hg in crops mention this as a potential uptake mechanism, this research has largely focussed on soil contamination as the primary source of crop exposure to Hg. Experimental design of future research should not discount soil uptake entirely, definitely not in the context of MeHg uptake in rice but assessment of the atmospheric Hg(0) concentration crops are exposed to should be an essential component of future studies in this area. Again, more accessible air monitoring technologies such as passive sampling are likely the most effective strategy considering that most ASGM happens in the Global South. Such data are not only critical for assessment crop exposures to atmospheric Hg, but also to assess the magnitude of ASGM emissions at specific sites (Szponar et al., 2025). As posited by Arrazy et al. (2024) and



Rothenberg et al. (2014) the types of ASGM activities and the intensity and age of those
activities as influencing factors on crop Hg concentrations and speciation.
The complexity of MeHg production, and paddy cycling of Hg, have been under appreciated
in ASGM environments. Including such analyses in future work would improve interpretation
of studies that observe anomalous data of low soil and high rice THg concentrations (and
vice versa). Future work should incorporate measurements of Hg(0) at the studied paddies
to assess atmospheric exposures of rice to Hg(0) and delineate the burden of THg in rice
coming from air-stomata uptake pathway (and potentially direct sorption of atmospheric Hg
species to developing grain). Adding measurements of MeHg in soil and grain compartments
would allow greater capacity to differentiate if anomalous high soil/low rice or low soil/high
rice THg concentrations are driven more by variable methylation rates in different paddies
or a greater fraction of THg in rice being derived from the Hg(0) stomatal assimilation
pathway than previously thought.
Authors focused on concentration data and seldom measured the biogeochemical factors
that could help explain and understand methylation in ASGM rice paddies. Data on relevant
soil and water biogeochemistry is limited to nearby waterways, rather than paddies
(Appleton et al. 2006 and Pataranawat et al. 2007). Where feasible, measurements of
methylation and demethylation rate potentials, Hg stable isotopes (or isotope
enrichments), and complementary biogeochemical analyses (i.e., pH, temperature, redox
conditions, carbon composition) are also needed. It is important to note that even if
methylation rates are low, the extremely high supplies of inorganic mercury in ASGM
environments can still lead to high concentrations of MeHg; this question remains largely
unexplored. These knowledge gaps of Hg cycling in ASGM impacted paddy soils limit our
capacity to identify specific drivers of elevated MeHg production and the associated health
risks. This in turn makes it difficult to identify which agricultural strategies that have
potential to reduce paddy production of MeHg and accumulation in rice grains (i.e., biochar
amendment, alternative wetting and drying cultivation, or the use of low-MeHg
accumulating cultivars; Tang et al., 2020).
We must consider that the range of crops potentially affected by ASGM activities is broad.
C3 and C4 plants have different photosynthetic pathways, which as Eboigbe et al. (2025)
speculate could lead to differing rates of Hg(0) uptake from air. Xia et al. (2020) suggest
longevity of crops (annuals vs perennials) may also impact Hg uptake rates from air and/or
soils. Future work should not only broaden the range of crop species exposed to Hg
contamination from ASGM, but also as many different crop tissues, beyond simply edible



parts, as possible, and even different compartments of individual tissues (i.e., tubers: peels
vs flesh; stems and roots: cortex/epidermis vs vascular bundles vs pith). Such detail is
crucial for subsurface tissues (i.e., roots) as it has been suggested that the root epidermis
is an effective barrier preventing uptake of inorganic Hg species (Lomonte et al., 2020).
Applying Hg stable isotope analyses to the different sections of dissected tissues has the
potential to identify the source of Hg in each tissue section using two end-member mixing
models for the air and soil uptake pathways (as applied in Eboigbe et al., 2025) as well as
elucidate information on the internal translocation of Hg by these crops. Development of a
process-based vegetation model examining internal Hg cycling using THg, Hg(0), IHg(II), and
MeHg concentrations and stable isotope (including fractionation factors) would be a major
advancement not just for ASGM impacted farming systems, but for all study of Hg in
vegetation.
It is a clear from our review that there is a dearth of information relating to Hg in livestock
and poultry meat and dairy/egg by-products be that in high-risk ASGM areas or otherwise.
Concerns of inorganic Hg bioaccumulation and health impacts are evidenced by Hg in
livestock and poultry, particularly in detoxifying organs like the kidney and livers. More study
required to understand the health risks to livestock/poultry themselves and humans
consuming them (and their edible by-products) through the examination of THg and MeHg
concentrations. Moreover, future work should better examine the transfer of Hg from
contaminated feedstocks to these animals and determine the role Hg speciation in
feedstocks plays in this transfer. Adding Hg stable isotopes to such assessments would
improve our mechanistic understanding of Hg uptake, cycling, and fate within animals
farmed in areas adjacent to ASGM.
Another important gap is that the effects of food preparation are not included in estimates
of daily intake. Understanding of the effects of cooking on Hg and MeHg bioavailability has
only recently coalesced, and is still limited to *in vitro* studies, which has been recently
reviewed by Gong et al. (2025). The bioaccessibility of THg and MeHg vary widely between
foodstuffs based on the macronutrient composition of food preparation methods (i.e.,
grinding vs. whole grain), , and cooking methods (high temperature cooking can reduce
MeHg bioaccessibility) (Gong et al., 2025). With this considered, it is essential that there be
a greater focus of research into the effects of meal composition and preparation and
cooking methods on Hg concentrations, speciation, and bioavailability in edible crop parts
and livestock and poultry meats and eggs/dairy. This is particularly so for areas impacted by
ASGM activities due to greater potential Hg exposure via contaminated foods.



Bridging these barriers will require multidisciplinary approaches involving collaboration with
mine stakeholders, community leaders and engaged citizens, and both local and
international scientists to conduct safe and effective site assays that effectively address the
critical knowledge gaps outlined in this work. As highlighted by Moreno-Brush et al. (2020),
we stress the importance of international collaborations between scientists in areas directly
impacted by ASGM that possess key local partnerships and knowledges of geographies,
customs, and cultures, and those from the Global North with access to greater funding
opportunities and advanced methodologies (i.e., Hg stable isotope instrumentation, global
fate and transport models) critical to generating the scientific robustness and impact
needed to assess the impacts of ASGM Hg use on terrestrial agricultural communities.
Equally vital is also ensuring knowledge translation to impacted communities post-research
by promoting respectful engagement, avoiding exploitation (parachute/colonial science),
and fostering lasting collaborations (Kukkonen and Copper, 2019). The production of
knowledge alone should not be the sole motivator in such efforts. Growth of ASGM is driven
by demand for gold in the Global North and rapidly developing economies in Asia (Verbrugge
and Geenen, 2020; Prescott et al., 2022); hence, there is responsibility that this global issue
(and its impacts) requires global solutions.

## Supplement and Code/Data Availability

There is no supplement or additional code/data associated with this literature review.

## Author Contributions

All authors contributed to the writing and reviewing of this work.

## Competing Interests

D.S.M. is a member of the editorial board of the journal *Biogeosciences*. The authors declare
that they have no other conflict of interest.

## Acknowledgements

As noted in Figure 1 caption, we acknowledge the use of ChatGPT (OpenAI) to create the
plant and plant tissue images. However, the figure text, labels, and arrangement was
completed by co-authors.



## Financial Support

We acknowledge an internal, alumni-based, graduate scholarship/award (Maria Nathanson/IAMGOLD Fund: 50540) and D.S.M.'s Research Initiation Grant, both from Queen's University, for funding this work.

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
