# Peer review of "Reviews and syntheses: Artisanal small-scale"

_EGUsphere, 2025_

## Author Comment (AC2)

**Reviews and syntheses: Artisanal small-scale gold mining (ASGM)-derived mercury contamination in agricultural systems: what we know and need to know**

**RESPONSE TO REVIEWERS:**

We note our responses are in blue and we use the notations RxCx to define a specifically numbered comment (C) relating to a specifically numbered reviewer (R). RxARx refers to a specifically numbered Author Response (AR) that relates to a reviewer comment.

**REVIEWER 1 (R1):**

**R1C1:** The article "Reviews and syntheses: Artisanal small-scale gold mining (ASGM)-derived mercury contamination in agricultural systems: what we know and need to know" focuses on crops grown at ASGM sites and livestock/poultry as potential sources of human exposure to Hg through ASGM agriculture, while omitting the extensively studied fish, which are covered by other authors. The related literature is reviewed, and relevant data is extracted and synthesized.

**General comments:**

The review/synthesis is thorough and easy to read. From my side, there are no major comments, which points to the overall good quality of the manuscript. Therefore, only specific comments and technical corrections are outlined below.

**R1AR1:** We thank the reviewer for their strong support of this work and for their time, dedication, and contribution they have made. It is greatly appreciated.

**Specific comments:**

**R1C2:** Section 3: This section serves as the opening to the core of the work done by the authors. It would benefit from the inclusion of a "motivation" part for this work, where the authors state related review/synthesis articles and explain how this work is different. This addition would save the reader some exploration and points to other related reviews should they need it.

How did the authors collect the data? Which search tools were used for finding relevant articles? What were the key words used in the search, and did the authors conduct the search with a specific Boolean search query? How were the decisions made regarding the inclusion/exclusion of data for the synthesis? As it stands, the article search/data collection is not traceable and reproducible. I recommend adding an extra short section dedicated to "methods" or "data collection", similar to e.g. Basu et al. (2018,

https://ehp.niehs.nih.gov/doi/10.1289/EHP3904). It is true that the amount of data collected for ASGM agricultural systems is much smaller than in Basu et al. (2018); however, some description would still benefit the article.

**R1AR2:** We appreciate the authors comment and suggestion for structure and reproducibility. However, we emphasize that this is a more traditional literature review structure and is not a systematic review. The two main reasons for not choosing a systematic literature review are as follows:

The literature on this topic is narrower in quantity (as the reviewer highlights) than say a
study on mercury biomarkers in human populations across the world (Basu et al).
Narrowing the selection of literature from a broader pool of literature is essentially the
purpose of a systematic approach. In our review, our search of the literature was
intended to be exhaustive; we did not (by choice) exclude studies. Any studies that were

left out of Table 1 or Table 2 were done so due data issues that could not be resolved through contact with the original authors or discussion with the group; yet these studies were still mentioned in the manuscript (i.e., lines 563-567).

All three authors searched databases (web of science and google scholar) and studies were collated. As different sections were written, the authors continued to search for new literature and if new literature was found it was added and shared with the group. The primary search terms were artisanal and small scale gold mining OR ASGM OR small scale mining AND mercury AND crops OR livestock OR poultry. However, more specific searches that included regions (i.e., Asia, South America, Africa) or specific crop or animal names were also used to ensure more generalized searches did not miss key studies. Individual authors were flexible to create their own, narrow searches.

2. Systematic reviews are structured by the *a priori* knowledge of the search team to test a specific hypothesis (Uttley et al., 2023). That again was not our purpose due to the limited scope of the literature and the complex, interdisciplinary nature of the ASGM problem (which has been our goal to highlight). We wanted to minimize the bias from our own *a priori* knowledge in shaping the our review of the literature in this subject area. We wanted our review of the literature to continually evolve the structure and discussion created in this more "traditional" literature review rather than being somewhat constrained by our original ideas and viewpoints. We believe searching individual and then bringing that together collectively was an effective means of executing that plan.

In terms of recognizing other reviews in somewhat similar topic areas, we do recognize these at numerous points within the manuscript (i.e., Lines 280-283, lines 660-662, Line 1088, Line 358-360). Indeed, we would suggest that our review follows that of other example reviews in the field such as those by Gačnik and Gustin (2023), Liu et al. (2022), and Zhou et al. (2021) – all of which are referenced in our work, are structured around narrative, and all of which do not include search description methods or statements.

Since, this is not a systematic review, we would prefer the start of section 3 remain as it is to allow the progression of the existing central narrative (which the reviewer is supportive of) rather than disrupt this with a search method description.

**R1C3:** Lines 81-83: Solubility is a physical rather than a chemical property. The authors could include chemical properties and list. in the first bracket, an actual chemical property (for example, the formation of different IHg(II) complexes with various inorganic/organic ligands) and then a physical property (here, the current solubility example can be kept).

**R1AR3:** We thank the reviewer for picking up this oversight, this should state "physicochemical properties" and will be corrected in the resubmitted version of the manuscript.

**R1C4:** Lines 88-97: The paragraph starts with an explanation of the global Hg character due to  $Hg^0$  properties, and understandably, the story then continues with a focus on  $Hg^0$ . However, it should not be left out that the emissions are not only in the form of  $Hg^0$  but also directly as IHg(II) (g) and IHg(II) on particulates. Currently, it reads as if the only way for Hg to deposit into terrestrial/aquatic environments is through the oxidation of  $Hg^0$  to IHg(II) (g) and IHg(II) on particulates, and subsequent deposition. In fact, IHg(II) (g) and IHg(II) on particulates can be emitted directly from the emission source and deposit locally without involvement of redox

processes, as I am sure the authors are well aware. This needs to be clarified in the text, if possible.

**R1AR4:** While we largely agree with the reviewer's assessment, we suggest that their assessment is based on industrial (combustion) emissions (i.e., industry/fossil fuels). We stress that ASGM emissions to air as being quite different from industrial combustion based emissions. Nonetheless, we have updated lines 104-108 to reflect these potential direct IHg(II) emissions to air:

"IHg(II) compounds deposited, produced in situ from Hg(0) oxidation, emitted directly as IHg(II) to air from some industrial source, or released directly into aquatic environments such as wetlands, rivers, and lakes can undergo microbially mediated (both enzymatic and non-enzymatic) processes that catalyse the transfer of methyl groups from donors like methylcobalamin to IHg(II) species, forming MeHg compounds (Ullrich et al., 2010)."

**R1C5:** Lines 166-168: The "note II" on the bottom of the page states, "Note the estimate of primary releases to aquatic systems does not include releases from ASGM activities as the...". On the other hand, in lines 166-168, numbers appear that estimate total ASGM releases to water and land, which is a bit contradictory to the note. So, the releases to land and water from ASGM were estimated, but water-only releases were not?

**R1AR7:** We have reviewed these estimates again (and we spent considerable time on this before the original submission) and the statements are correct. Releases to land and water are estimated combined, but a water alone estimate was not made. We acknowledge that there is some ambiguity here so we tried to reflect the actual reporting as closely as possible (including the use of the note). All this information was relayed to stress out poorly constrained all ASGM emissions (air) and released (land and water) are. This is a highly uncertain "special case".

**Technical corrections:**

R1C6: Line 294: "operate" should likely be "operating", or "operation"

R1AR6: This will be changed to "to operate".

R1C7: Line 607: Should be "facilitates oxide regeneration of sulphate ... "

**R1AR7:** This should remain as is. It is the oxic conditions created by paddy draining that regenerates sulphate and ferric iron via both microbial processes and abiotic oxidation. We feel that "oxic conditions" captures both aspects of the biogeochemistry.

**R1C8:** Lines 639-643: The paragraph sounds like MeHg is only demethylated in foliage, but later on, a more general "in-planta" demethylation is referred to. Can this be clarified?

**R1AR8:** We greatly appreciate this comment from the reviewer. The exact location of in-planta demethylation within tissues is not known. We have adjusted the writing at lines 639-643 to refer to "aboveground tissues" generally.

R1C9: Line 647: Is "#orava" a typo? Otherwise, not clear what it should mean

**R1AR9:** Thanks for picking this up, it is was a placeholder used by one of the authors that was never deleted. It will be deleted in the revised manuscript.

R1C10: Line 657: "sorbed" should be "sorb"

R1AR10: Will be corrected.

R1C11: Line 693: There is a typo at the end of the sentence, "alter estimates."

**R1AR11:** We believe this was a hangover from an earlier edit. This will be deleted.

R1C12: Line 848: "... in SE Asia, simply that ..." there should be a "and" / "or" after the comma

R1AR12: "or" will be added

R1C13: Line 959: Missing "in" in this part of the sentence: "7.3x higher muscle"

R1AR13: Will be fixed.

R1C14: Line 992: "high" should be "higher"

R1AR14: Will be fixed.

R1C15: Line 1090: There is a redundant comma in the sentence

R1AR15: Will be deleted.

REVIEWER 2 (R2):

General comments:

**R2C1:** The review by McLagan et al. is excellently written and covers a novel topic of interest: the role of ASGM in agricultural systems. The manuscript represents a useful resource for readers interested in the available literature on the specific subject and the critical gaps remaining to be investigated, as well as covering the broader aspects of Hg uptake by vegetation. The literature cited is extensive and the review is well-organized. I commend the authors on their work and offer several minor comments and suggestions for improvement.

**R2AR1:** Again we greatly appreciate the support and kind sentiments as well as the time that have given to provide this contribution. It is greatly appreciated.

Specific comments:

**R2C2:** At several points in the manuscript (L464–468, L502–506), the authors suggest a correlation between soil Hg levels and atmospheric Hg0 levels to substitute for the fact that air measurements were not available from certain studies. I am sceptical whether this relationship holds for areas close to ASGM and agricultural activities, as these activities cause deforestation that can lead to release of Hg from soils (more erosion, less Hg0 uptake, and more Hg0 volatilization from soils). For example, see Figure 2B,E from Gerson et al. 2022 (doi:10.1038/s41467-022-27997-3), where you can see that for mining impacted sites there is not necessarily a strong correlation between soil Hg and GEM. There is an array of confounding factors that likely affect the relationship between air GEM and soil Hg concentrations, including the mining techniques, time since deforestation and the soil biogeochemical factors. For example, Carpi et al. (doi:10.1016/j.atmosenv.2014.08.004, 2014) reported that a freshly deforested site showed higher soil Hg concentrations than a pasture site that was deforested 10 years earlier. All this to say that these confounding factors could explain why the air Hg levels, and hence the THg levels in crops, do not always follow the soil concentrations.

**R2AR3:** This commented is well noted, thank you. We did attempt to describe the uncertainty this assumption creates ("While less ideal than paired soils and atmosphere measurement..."). However, on reflection we realize this is suboptimal. We will add the following statement at the end of the first paragraph flagged by reviewer 2:

"However, we acknowledge that there can be exceptions to this relationship including in ASGM areas (Gerson et al., 2022); and hence acknowledge the elevated uncertainty such an assumption creates."

**R2C3:** I noticed a couple of instances of discrepancies between years of in-text references vs. bibliography (Zhou and Obrist, 2022 on L393 vs. 2021 in bibliography) and missing references (L406 Yanai et al. 2020). Perhaps worth double checking in case more references were missing from bibliography.

**R2AR3:** We greatly appreciate the level of detail the reviewer has taken here. These changes will be made. Due to the high number of references, we had run the citations and references through several human and automated checks. We will repeat this again before submission of the revised manuscript and hope that we will have picked up all the outstanding issues.

R2C4: L157 - Worthwhile mentioning UNEP estimate for releases to land as well?

**R2AR4:** These data are not directly reported as summative values and are challenging to calculate due to little information of releases to waste streams and releases to land from many sectors. Hence we cannot add this to listed estimates.

**R2C5:** L334 - As part of the stomatal vs. non-stomatal discussion, might want to include this recent study which analyzed this issue in a tropical rainforest greenhouse, finding clear evidence of dominance of the stomatal pathway: Denzler et al., doi:10.1021/acs.est.5c05823, 2025

**R2AR5:** This is a really valuable reference (thank you) and fits perfectly alongside the Naharro et al., 2020 reference (and has been added to the revised manuscript).

**R2C6:** L420 - A recent study from Tibet discussed the issue of root vs. atmosphere uptake found a relationship with altitude (Wang et al., doi:10.1038/s43247-022-00619-y, 2022), but perhaps less relevant for ASGM, nonpermafrost regions

**R2AR6:** This does seem like a very interesting paper, but it is quite a niche study assessing highaltitude tundra/permafrost. Hence, we tend to agree with the reviewer that relevancy is too large of an issue for applicability to this work.

Technical comments:

**R2C7:** Title - minor, but I've usually seen ASGM defined with an "and" - Artisanal and small-scale gold mining

**R2AR7:** Minor, but still important and it was not consistent throughout. We have standardised all use in the revised manuscript to include "and"

R2C8: L12 - systems (plural?)

R2AR8: Will be Corrected.

**R2C9:** L230- is the 60% by mass?

R2AR9: "by mass" will be added.

R2C10: L382 - rephrase to amend unclear "their"

**R2AR10:** Rephrased to make the authors the subject, which we believe clarifies "their". "Using a Hg stable isotope mass balance model, Yuan et al. (2018) estimated that ≈30% of assimilated Hg(0) is re-released from their studied species."

**R2C11:** L424 - linked the uptake (missing "to")

R2AR11: Will be corrected.

R2C12: L440 - missing comma before highlighting

**R2AR12:** We do not believe a comma is required here.

**R2C13:** L441 - missing "the" before fraction

**R2AR13:** Will be corrected along with some other erroneous text that required updating after recalculations were made in the Eboigbe et al., 2025 paper before it was accepted for publication.

R2C14: L467 - check this assumption, does air Hg0 actually correlate well with soil THg?

**R2AR14:** As per previous comment an additional sentence was added to this paragraph to clarify this point and the uncertainty of this assumption. Ultimately, we'd prefer not to make it, but many studies lack atmospheric Hg measurements.

R2C15: L490 - delete second at

R2AR15: Will be corrected.

**R2C16:** L517-521 - this sentence meaning is unclear to me (what does "summative" data mean here); potential this additional reference that is not considered should just be deleted for conciseness

**R2AR16:** What we mean is that concentration data from two different parts of a plant cannot be added together to give the THg concentration for the combined tissues (weighted averages must be calculated). We emphasize that there is some challenges with data reporting in some of the examined studies. However, we want to reiterate that our goal was to be exhaustive in listing all relevant studies in sections 3.1, 3.2, and 3.3, and we want to make sure that we do not present the appearance of arbitrarily omitting studies, particularly as a big challenge in doing this work is access to scientific resources, which are bias to the Global North. Hence, we believe it is best to leave this study and accompany it with the current structure.

**R2C17:** Table 1 - Arrazy et al. (2023) studied Indonesian site (also according to text), but listed is Tanzania as location

R2AR17: Will be corrected.

R2C18: L555 - a village set up around

R2AR18: Will be corrected.

**R2C19:** L642 - this number is missing a currency

R2AR19: US\$ will be added

**R2C20:** L647 - #orava typo

**R2AR20:** Thanks for picking this up, it is was a placeholder used by one of the authors that was never deleted. It will be deleted in the revised manuscript.

R2C21: L855 - please specify: higher soil THg levels

R2AR21: Will be corrected to "THg concentrations in paddy soils"

R2C22: L877 - was briefly discussed

R2AR22: Will be corrected.

**R2C23:** Table 2 - I'm assuming that asterisks mean estimates based on assumptions, but this should be specified in the caption or a table footnote

R2AR23: Will be corrected.

R2C24: L1022- comma missing before but

**R2AR24:** We believe this is the correct non-use of a comma (associated with but).

R2C25: L1071 - isotopes (plural)

R2AR25: Will be corrected.

R2C26: L1077-1080 - this sentence is missing a verb

R2AR26: Will be corrected to include "is".

**REFERENCES:**

Gačnik, J. and Gustin, M.S., 2023. Tree rings as historical archives of atmospheric mercury: A critical review. *Science of The Total Environment*, 898, p.165562.

Liu, Y., Liu, G., Wang, Z., Guo, Y., Yin, Y., Zhang, X., Cai, Y. and Jiang, G., 2022. Understanding foliar accumulation of atmospheric Hg in terrestrial vegetation: progress and challenges. *Critical Reviews in Environmental Science and Technology*, *52*(24), pp.4331-4352.

Uttley, L., Quintana, D.S., Montgomery, P., Carroll, C., Page, M.J., Falzon, L., Sutton, A. and Moher, D., 2023. The problems with systematic reviews: a living systematic review. *Journal of Clinical Epidemiology*, 156, pp.30-41.

Zhou, J., Obrist, D., Dastoor, A., Jiskra, M. and Ryjkov, A., 2021. Vegetation uptake of mercury and impacts on global cycling. *Nature Reviews Earth & Environment*, *2*(4), pp.269-284.